# Unusual facet and co-catalyst effects in TiO$_2$-based photocatalytic coupling of methane

Huizhen Zhang[1,3], Pengfei Sun[1,3], Xiaozhen Fei[1], Xuejiao Wu[1], Zongyi Huang[1], Wanfu Zhong[1], Qiaobin Gong[1], Yanping Zheng[1], Qinghong Zhang [1], Shunji Xie[1,2] ✉, Gang Fu [1,2] ✉ & Ye Wang [1,2] ✉

Photocatalytic coupling of methane to ethane and ethylene (C$_2$ compounds) offers a promising approach to utilizing the abundant methane resource. However, the state-of-the-art photocatalysts usually suffer from very limited C$_2$ formation rates. Here, we report our discovery that the anatase TiO$_2$ nanocrystals mainly exposing {101} facets, which are generally considered less active in photocatalysis, demonstrate surprisingly better performances than those exposing the high-energy {001} facet. The palladium co-catalyst plays a pivotal role and the Pd$^{2+}$ site on co-catalyst accounts for the selective C$_2$ formation. We unveil that the anatase {101} facet favors the formation of hydroxyl radicals in aqueous phase near the surface, where they activate methane molecules into methyl radicals, and the Pd$^{2+}$ site participates in facilitating the adsorption and coupling of methyl radicals. This work provides a strategy to design efficient nanocatalysts for selective photocatalytic methane coupling by reaction-space separation to optimize heterogeneous-homogeneous reactions at solid-liquid interfaces.

As the major constituent of natural gas, shale gas, biogas, and combustible ice, methane has attracted much attention for the production of value-added chemicals and easily transportable fuels. The current industrial process for the chemical utilization of methane is limited to reforming to offer syngas (CO + H$_2$), followed by syngas transformations. As compared to this indirect route, a direct methane conversion route would be more energy- and cost-efficient, and the development of the direct methane conversion route has long been regarded as one of the most fascinating research goals[1]. Among various possible methane valorization reactions, the non-oxidative coupling of methane (NOCM) to C$_2$ compounds (ethane and ethylene) is of particular interest because of the simultaneous production of valuable H$_2$ and the potential zero emission of CO$_2$[2,3]. However, the NOCM to either ethane or ethylene is a highly endothermic reaction and is thermodynamically limited under mild conditions. A high reaction temperature (> 650 °C) is required to trigger the NOCM by thermocatalysis,

usually leading to deep dehydrogenative by-products or even carbon deposition[4].

Photocatalysis, driven by solar energy, provides a promising way for the transformation of methane to high-value chemicals under mild conditions[5–13]. Many recent studies have been devoted to developing the NOCM photocatalysts[14], and metal oxide (e.g., TiO$_2$, ZnO, and TiO$_2$ − SiO$_2$)-based semiconductors loaded or doped with various co-catalysts have shown promising performances for the formation of C$_2$ compounds[15–21]. It has been proposed that the photogenerated hole centers on oxide surfaces (O$^{2-}$ + h$^+$ ⟶ O$^-$) or hydroxyl radicals (•OH) in the presence of H$_2$O may be responsible for activating the C−H bond in CH$_4$ under mild conditions, forming •CH$_3$ for C−C coupling to C$_2$H$_6$ as the major product[13]. However, these active oxygen species with strong oxidation ability are also known to cause uncontrollable oxidation reactions[12,13,22], lowering the formation rate of C$_2$ compounds[12–14]. As a result, the C$_2$ formation rate can hardly exceed

[1]State Key Laboratory of Physical Chemistry of Solid Surfaces, Collaborative Innovation Center of Chemistry for Energy Materials, National Engineering Laboratory for Green Chemical Productions of Alcohols, Ethers and Esters, College of Chemistry and Chemical Engineering, Xiamen University, Xiamen, China. [2]Innovation Laboratory for Sciences and Technologies of Energy Materials of Fujian Province (IKKEM), Xiamen, China. [3]These authors contributed equally: Huizhen Zhang, Pengfei Sun. ✉e-mail: shunji_xie@xmu.edu.cn; gfu@xmu.edu.cn; wangye@xmu.edu.cn

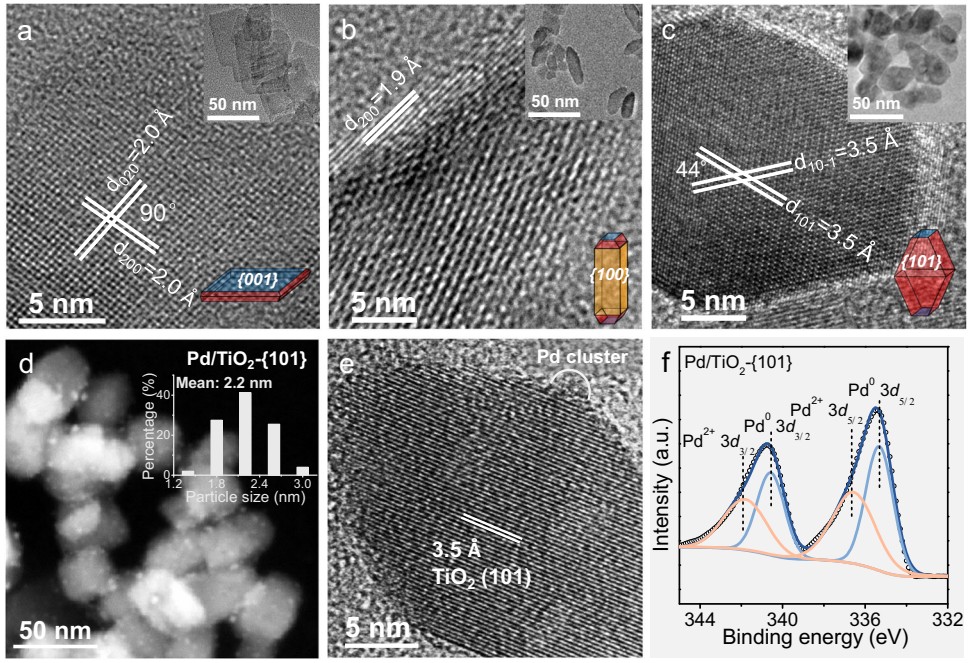

**Fig. 1 | Catalyst morphologies and structures. a** HRTEM and TEM (inset) images of TiO$_2$-{001}. **b** HRTEM and TEM (inset) images of TiO$_2$-{100}. **c** HRTEM and TEM (inset) images of TiO$_2$-{101}. **d** STEM image and Pd particle size distribution (inset) of Pd/TiO$_2$-{101}. **e** HRTEM image of Pd/TiO$_2$-{101}. **f** Pd 3$d$ XPS spectrum of Pd/TiO$_2$-{101}.

100 μmol g$^{-1}$ h$^{-1}$ over most of the photocatalysts reported to date[14]. It is highly attractive and challenging to develop useful strategies for the design and construction of efficient photocatalysts to increase the efficiency of the NOCM.

The loading of suitable co-catalysts and the manipulation of the nanostructure of a semiconductor photocatalyst are two key strategies to improve photocatalytic performance. Some noble metal co-catalysts such as Pt[16], Ag[18], Au[19], and Pd[20,21] have been exploited for the metal oxide-based photocatalytic NOCM. For example, a single-atom Pd-modified TiO$_2$ catalyst (Pd$_1$/TiO$_2$) showed a high activity among different catalysts (Supplementary Table 1), and the accumulated photogenerated holes on Pd sites were proposed for the activation and conversion of CH$_4$ to C$_2$H$_6$[21]. Engineering the exposed facets of semiconductor nanocrystals is another useful strategy to tune the photocatalytic performance. The current consensus is that the facet with higher surface energy is more active in photocatalytic reactions[23,24]. As a representative example, the anatase TiO$_2$ {001} facet with a higher surface energy has been proven to possess higher photocatalytic activity than the {101} facet with a lower surface energy[25-27]. Recently, such a facet effect has also been proven in photocatalytic CH$_4$ conversions[28,29]. The {001}-dominated TiO$_2$ showed a significantly higher CH$_3$OH formation rate than the {101}-dominated TiO$_2$ in photocatalytic CH$_4$ oxidation by O$_2$[28], confirming the role of high-energy surfaces. For the photocatalytic NOCM, a polar {001}-dominated ZnO was similarly beneficial to CH$_4$ activation and C$_2$H$_6$ formation[29], offering a C$_2$H$_6$ formation rate of about 10 μmol g$^{-1}$ h$^{-1}$.

In the present work, we report our discovery of an unusual phenomenon that the anatase TiO$_2$ nanocrystal mainly exposing the stable {101} facet shows the best performance, whereas that with the high-energy {001} facet is significantly less active and selective for the photocatalytic NOCM in the presence of water. The Pd co-catalyst also plays a crucial role in C$_2$ formation, and it is unique that the positively charged Pd species, rather than the metallic Pd species accounts for the selective formation of C$_2$ compounds. We achieve a CH$_4$ conversion rate of 326 μmol g$^{-1}$ h$^{-1}$ with 81% selectivity of C$_2$ compounds over {101}-dominated anatase TiO$_2$ nanocrystals modified by Pd$^{2+}$-containing co-catalyst, and the attained C$_2$ formation rate represents one of the best values reported to date (Supplementary Table 1). Our studies

unveil that this unusual finding is related to the location of the active species generated in photocatalysis. The •OH radicals formed in the liquid phase are proposed to work as the active species for the selective conversion of CH$_4$ to C$_2$H$_6$, whereas those strongly adsorbed on the surface (in the case of {001}-dominant TiO$_2$) cannot efficiently participate in CH$_4$ activation or may lead to over-oxidation. Our studies further suggest that the Pd co-catalyst contributes to aiding the coupling of •CH$_3$ radicals by enhancing their adsorption onto catalyst surfaces, besides enhancing the electron-hole separation.

## Results

### Characterizations of Pd-loaded TiO$_2$ nanocrystals

Three types of TiO$_2$ nanocrystals with different morphologies and exposed facets were synthesized by hydrothermal synthesis[30]. Transmission electron microscopy (TEM) observations show that the obtained nanocrystals are in nanosheet, nanorod, and nanobipyramid morphologies (Supplementary Fig. 1a–c). High-resolution TEM (HRTEM) reveals that the TiO$_2$ samples with nanosheet, nanorod, and nanobipyramid morphologies mainly expose facets of {001}, {100}, and {101}, respectively (Figs. 1a-1c). A detailed analysis of the TEM result indicates that the fraction of the mainly exposed facet exceeds 80% in each TiO$_2$ sample (Supplementary Fig. 2). Hereafter, we denote these three nanocrystals as TiO$_2$-{001}, TiO$_2$-{100}, and TiO$_2$-{101} for brevity. X-ray diffraction (XRD) and Raman spectroscopy both reveal that the three TiO$_2$ nanocrystals are in the anatase phase (Supplementary Fig. 3a and 3b). The difference in the broadening of XRD peaks probably arises from the difference in the average sizes of TiO$_2$ nanocrystals. Compared to TiO$_2$-{101} and TiO$_2$-{100}, the TiO$_2$-{001} sample showed relatively stronger (200) and weaker (004) diffraction peaks (Supplementary Fig. 3a), in agreement with the TEM result that the TiO$_2$-{001} sample has the largest side length in the [100] direction and the smallest thickness in the [001] direction[31]. The $E_g$ bands at 144 and 636 cm$^{-1}$ in Raman spectra are attributable to the symmetric stretching vibration of O−Ti−O of TiO$_2$ (Ref. [31]), and the weaker $E_g$ bands for TiO$_2$-{001} (Supplementary Fig. 3b) are consistent with the fact that this sample exposes a higher fraction of {001} facet with fewer symmetric stretching vibration modes of O−Ti−O. X-ray photoelectron spectroscopy (XPS) measurements confirm that the TiO$_2$ nanocrystals are

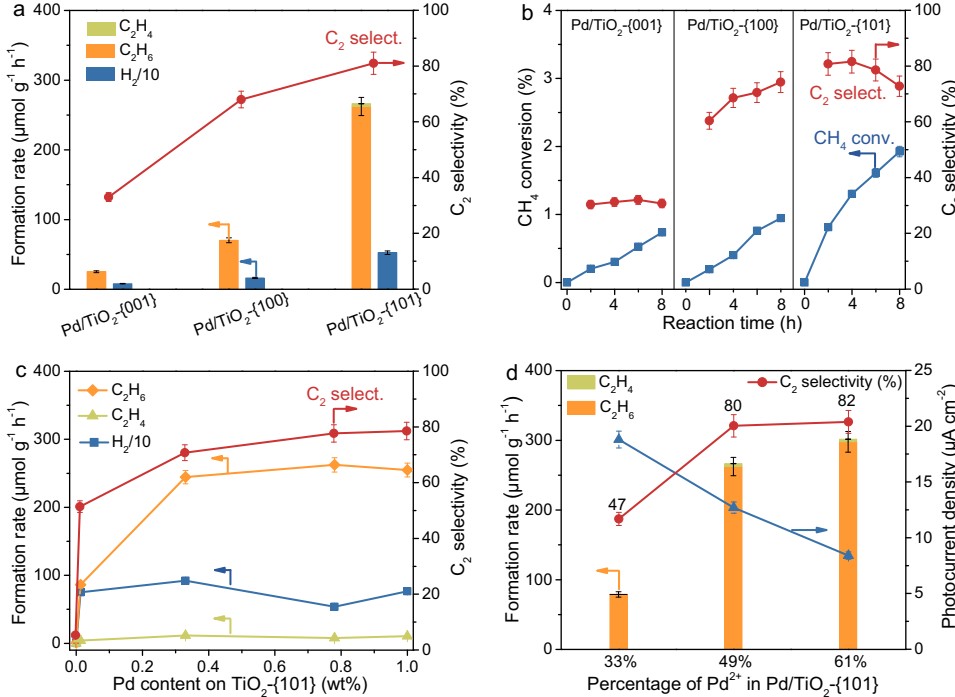

**Fig. 2 | Photocatalytic behaviors. a** Effect of exposed anatase facet on NOCM performances over Pd/TiO₂ catalysts. **b** Time courses for CH₄ conversion and C₂ selectivity over Pd/TiO₂ catalysts with different exposed anatase facets. **c** Effect of Pd content on NOCM performances over Pd/TiO₂-{101} catalysts. **d** Effect of the percentages of Pd²⁺ on Pd nanoparticles on NOCM performances and transient

photocurrent densities over Pd/TiO₂-{101} catalysts. Reaction conditions: catalyst, 0.020 g; water, 50 mL; CH₄, 45 mL (2009 μmol); light source, 300 W Xe lamp ($\lambda = 320$–780 nm). The error bar represents the relative deviation, which is within 5%.

composed of only Ti and O elements without other impurities and that the oxidation states of Ti or O are the same in the three samples (Supplementary Fig. 3c–3e).

Pd species were loaded onto TiO₂ by an adsorption-reduction method and the measured contents of Pd loaded on TiO₂-{001}, TiO₂-{100}, and TiO₂-{101} were almost the same (Supplementary Table 2). The morphologies and the crystalline structures of the three TiO₂ samples did not change after loading Pd (Supplementary Fig. 1d–f and Supplementary Fig. 3f). The scanning transmission electron microscopy (STEM) showed that the average sizes of Pd loaded on the TiO₂-{101}, TiO₂-{100}, and TiO₂-{001} were 2.2–2.5 nm (Fig. 1d and Supplementary Fig. 4a, 4b). The HRTEM observation implies the noncrystalline cluster feature of the loaded Pd (Fig. 1e). Broad Pd 3*d* XPS spectra, which could be deconvoluted into two components of Pd⁰ and Pd²⁺ (Ref. 32), were observed over the three TiO₂ surfaces (Fig. 1f and Supplementary Fig. 4c, 4d). The fractions of Pd²⁺ estimated from the XPS spectra were also similar and close to 50% over the three Pd/TiO₂ samples (Supplementary Table 2). Furthermore, the fractions of Pd²⁺ on the Pd/TiO₂ catalysts basically kept unchanged after 4 h of photocatalytic NOCM reaction (Supplementary Fig. 5 and Supplementary Table 3). The high fraction of Pd²⁺ on our Pd/TiO₂ catalysts and the keeping of Pd²⁺ during the photocatalytic reaction could be attributed to the strong interaction between the small Pd nanoparticles or nanoclusters and TiO₂[33,34].

## Photocatalytic properties of Pd-loaded TiO₂ nanocrystals

The photocatalytic CH₄ conversion was performed in a batch reactor in the presence of H₂O at ambient temperature and pressure, and C₂H₆, C₂H₄, CO₂, and H₂ were detected over our Pd/TiO₂ catalysts. The performance was found to depend strongly on the exposed facets of TiO₂; the Pd/TiO₂-{101} catalyst demonstrated the best performance (Fig. 2a). Over this catalyst, the amounts of C₂H₆, C₂H₄, CO₂, and H₂ formed after 4 h of irradiation were 10, 0.16, 4.8, and 42 μmol,

respectively (Supplementary Table 4). The ratio of the amount of holes consumed in the formation of C₂H₆, C₂H₄, and CO₂ via CH₄ oxidation to that of electrons consumed in the formation of H₂ was calculated to be close to 1.0, confirming that the photogenerated hole and electrons participate in the formation of carbon-containing products and H₂, respectively. Control experiments showed no formation of carbon-containing products and a significantly decreased amount of H₂ by using N₂ instead of CH₄ (Supplementary Table 5). We only detected a trace amount of O₂ in the absence of CH₄, suggesting that the oxidation of H₂O to O₂ by photogenerated holes proceeded much slower, and this also decelerated the half-reaction of H₂ evolution by photogenerated electrons. No products were detected without light irradiation or in the absence of a catalyst. All these facts verify that the carbon-containing products originate from CH₄ via light-driven photocatalytic reactions.

A more detailed analysis shows that the Pd/TiO₂-{101} catalyst, which shows the best performance, offers a C₂H₆ formation rate of 262 μmol g⁻¹ h⁻¹, which is 3.7 and 11 times larger than those for the Pd/TiO₂-{100} and Pd/TiO₂-{001} catalysts (Fig. 2a). The C₂ selectivity is also the highest over the Pd/TiO₂-{101} catalyst (81%) and decreases in the sequence of Pd/TiO₂-{101} > Pd/TiO₂-{100} > Pd/TiO₂-{001}. The CH₄ conversion increased almost linearly with the reaction time over each catalyst, suggesting that the reaction proceeded steadily, and the values were 1.9%, 0.94%, and 0.75% over the Pd/TiO₂-{101}, Pd/TiO₂-{100}, and Pd/TiO₂-{001} catalysts, respectively, after 8 h of reaction (Fig. 2b). Meanwhile, the C₂ selectivity only changed slightly with the reaction time and the values over the Pd/TiO₂-{101}, Pd/TiO₂-{100}, and Pd/TiO₂-{001} catalysts remained at about 80%, 70%, and 30%, respectively, indicating that the C₂ products and CO₂ were formed in parallel from CH₄. We evaluated the specific C₂ formation rate on the basis of the surface area of TiO₂, and the values were 5.3, 0.74, and 0.37 μmol m⁻² h⁻¹ for the Pd/TiO₂-{101}, Pd/TiO₂-{100}, and Pd/TiO₂-{001}, respectively (Supplementary Table 6). Further, we have

compared the photocatalytic NOCM performance of the Pd/TiO$_2$-{101} catalyst with that of a Pd-promoted commercial P25 catalyst, which is known to possess high efficiencies in photocatalysis. The C$_2$ selectivity and yield of our Pd/TiO$_2$-{101} catalyst are significantly higher than those of Pd/P25 (Supplementary Table 4). These results demonstrate the intrinsic superiority of the {101} facet of TiO$_2$ in photocatalytic NOCM. Therefore, the TiO$_2$-catalyzed NOCM is a strong structure-sensitive reaction, and the exposed facet of TiO$_2$ is not only the CH$_4$ conversion but also the selectivity of C$_2$ compounds. We have measured the apparent quantum yield (AQY) of the Pd/TiO$_2$-{101} catalyst for the formation of C$_2$H$_6$ and C$_2$H$_4$, and the AQY value at a wavelength of 365 nm is 0.21%.

Our studies reveal that the Pd co-catalyst also plays a crucial role in accelerating the formation of C$_2$ compounds. In the absence of Pd co-catalyst, the formation rates of C$_2$ compounds and H$_2$ were both very low. The presence of Pd even with a very low content (0.013 wt%) could remarkably accelerate the formation of either C$_2$ compounds or H$_2$, and the C$_2$ selectivity was also significantly enhanced at the same time (Fig. 2c). A further increase in the Pd content up to 0.78 wt% increased the C$_{2+}$ selectivity and formation rate, although the H$_2$ formation was not significantly accelerated (Fig. 2c). It is noteworthy that the Pd/TiO$_2$-{101} catalyst displays a higher C$_2$ formation rate and C$_2$ selectivity than the Pd/TiO$_2$-{100} and Pd/TiO$_2$-{001} catalysts irrespective of the Pd content (Fig. 2c and Supplementary Fig. 6). We prepared Cl$^-$-free Pd/TiO$_2$-{101} catalyst by using Pd(NO$_3$)$_2$ as the Pd precursor instead of PdCl$_2$, which was typically used in the present work, and the XPS measurement confirmed the absence of chlorine on the surface of this catalyst (Supplementary Fig. 7). The Pd/TiO$_2$-{101} and the Cl$^-$-free Pd/TiO$_2$-{101} catalysts exhibited similar performances for photocatalytic NOCM (Supplementary Table 4), excluding the effect of Cl$^-$ residues. The decoration of TiO$_2$-{101} with other noble metal or transition metal co-catalysts including Pt, Au, Ag, Cu, or Fe could also promote the formation of C$_2$ compounds, but these co-catalysts except for Pt, showed remarkably poorer performances than Pd (Supplementary Fig. 8a). Pt was another highly efficient co-catalyst for C$_2$ formation, and the enhancing effect of Pt was only slightly inferior to that of Pd. The comparison of Pt-decorated TiO$_2$ catalysts with different exposed facets shows that the formation rate and selectivity of C$_2$ compounds decrease in the same sequence with the Pd-decorated series of catalysts, i.e., Pt/TiO$_2$-{101} > Pt/TiO$_2$-{100} > Pt/TiO$_2$-{001} (Supplementary Fig. 8b). This demonstrates once again that the TiO$_2$ {101} facet, which is the most stable and has been regarded as the least active facet in photocatalysis[25–27], shows the promising performance for the photocatalytic NOCM in the presence of a suitable co-catalyst.

We found that the oxidation state of the Pd co-catalyst exerted a significant effect on the photocatalytic NOCM. The method for post-treating the catalyst precursor, i.e., PdCl$_2$ adsorbed on TiO$_2$, could regulate the Pd oxidation state but keep the Pd content the same in the final catalyst. The H$_2$ reduction of the calcined PdCl$_2$/TiO$_2$-{101} precursor offered the Pd/TiO$_2$-{101} catalyst, which is typically adopted in the present work and has a surface Pd$^{2+}$/Pd$^0$ ratio close to 1.0 (Supplementary Table 2). On the other hand, the heat treatment in air and the reduction with NaBH$_4$ of the PdCl$_2$/TiO$_2$-{101} precursor offered catalysts (denoted as Pd-air/TiO$_2$-{101} and Pd-NaBH$_4$/TiO$_2$-{101}) with larger and smaller surface Pd$^{2+}$/Pd$^0$ ratios, respectively (Supplementary Fig. 9). The fractions of Pd$^{2+}$ on the surfaces of Pd-air/TiO$_2$-{101}, Pd/TiO$_2$-{101}, and Pd-NaBH$_4$/TiO$_2$-{101} were evaluated to be 61%, 49%, and 33%, respectively by XPS (Supplementary Fig. 9). The catalytic study reveals that a higher surface fraction of Pd$^{2+}$ results in a higher C$_2$ formation rate and higher C$_2$ selectivity (Fig. 2d). It is well known that the noble metal co-catalyst can accelerate the charge separation by accepting the photogenerated electrons from the semiconductor[35]. Our transient photocurrent response measurements showed that the photocurrent density decreased with an increase in the fraction of Pd$^{2+}$

(Fig. 2d and Supplementary Fig. 10a), indicating that Pd$^0$ plays a crucial role in accelerating the electron-hole separation. The control experiment for H$_2$ evolution in the presence of CH$_3$OH as a sacrificial agent showed that the H$_2$ evolution decreased upon increasing the fraction of Pd$^{2+}$ (Supplementary Fig. 10b). Therefore, we propose that Pd$^0$ accelerated the electron-hole separation and is responsible for H$_2$ formation, whereas Pd$^{2+}$ accounts for the formation of C$_2$ compounds in our system.

## Structure-property relationship and reaction mechanism

Our characterizations show that there are no significant differences in the light absorption and the energy-band structure among the three Pd/TiO$_2$ catalysts with different exposed facets (Fig. 3a and Supplementary Fig. 11). The electron-hole separation abilities of the Pd/TiO$_2$-{101} and Pd/TiO$_2$-{001} catalysts are also similar, although such ability of the Pd/TiO$_2$-{100} catalyst is lower (inset of Fig. 3a and Supplementary Fig. 12a). The photocatalytic H$_2$ evolution decreased in the sequence of Pd/TiO$_2$-{001} > Pd/TiO$_2$-{101} > Pd/TiO$_2$-{100} (Supplementary Fig. 12b). This trend of H$_2$ evolution is different from that for photocatalytic NOCM and could be explained by the higher electron-hole separation abilities of Pd/TiO$_2$-{001} and Pd/TiO$_2$-{101} (inset of Fig. 3a) and the higher surface energy of TiO$_2$-(001)[25]. The density of oxygen vacancies, which might determine the surface reactivity, was measured quantitatively by electron titration[27], and it followed a trend of Pd/TiO$_2$-{001} > Pd/TiO$_2$-{101} > Pd/TiO$_2$-{100} (Supplementary Fig. 13). This trend did not correlate well with that for the specific C$_2$H$_6$ formation rate and C$_2$ selectivity. Further, the surface area of the Pd/TiO$_2$-{101} catalyst with the best C$_2$ formation performance is the lowest (Supplementary Table 6). Therefore, it is not the shape-based parameters that may affect the light absorption, the electron-hole separation, and the surface area or the density of oxygen vacancies, but the exposed facet-based factor (not just the surface energy) plays a determining role in the TiO$_2$-catalysed NOCM.

Our catalytic result already implied that photogenerated holes and electrons functioned for the oxidative conversion of CH$_4$ and the reductive formation of H$_2$, respectively. We further found the crucial role of the presence of H$_2$O in the formation of C$_2$ compounds; the C$_2$ formation rate over the Pd/TiO$_2$-{101} catalyst was very low without H$_2$O and the presence of H$_2$O increased the C$_2$ formation rate for ~29 times (Supplementary Fig. 14). The substitution of H$_2$O by dimethyl sulfoxide (DMSO), N,N-dimethylformamide (DMF), or perfluorohexanes as the solvent resulted in no formation of C$_2$ products, further pointing out the determining role of H$_2$O. When terephthalic acid (25 μmol), a scavenger of •OH radicals[36,37], was added to the photocatalytic system with the Pd/TiO$_2$-{101} catalyst, the amounts of C$_2$H$_6$, C$_2$H$_4$, and CO$_2$ formed after 2 h of irradiation decreased drastically from 6.4, 0.16, and 3.1 μmol to 0.12, 0, and 0.28 μmol, respectively. We propose that the •OH radical formed in the presence of H$_2$O may participate in the activation of CH$_4$ and the formation of C$_2$ compounds.

The electron spin resonance (ESR) spectroscopic studies using 5,5-dimethyl-1-pyrroline N-oxide (DMPO) as a radical trapping agent showed a quartet spectrum corresponding to a DMPO-OH spin adduct[38], under photocatalytic conditions, confirming the generation of •OH radicals (Fig. 3b). The fluorescence measurements using terephthalic acid as a probe molecule could quantify •OH radicals in the liquid phase (Supplementary Fig. 15a-c and Fig. 3c)[36,37]. Both the ESR and the fluorescence measurements reveal that the concentration of •OH radicals in the liquid phase decreases in the sequence of Pd/TiO$_2$-{101} > Pd/TiO$_2$-{100} > Pd/TiO$_2$-{001}. This trend correlates well with that for C$_2$H$_6$ formation rate (Fig. 3c). We further performed surface fluorination to measure the total amount of •OH radicals generated (Supplementary Fig. 15d-f and inset of Fig. 3d), considering that •OH radicals may either be adsorbed on catalyst surfaces or exist in liquid phase and the addition of fluoride ions could release the •OH adsorbed

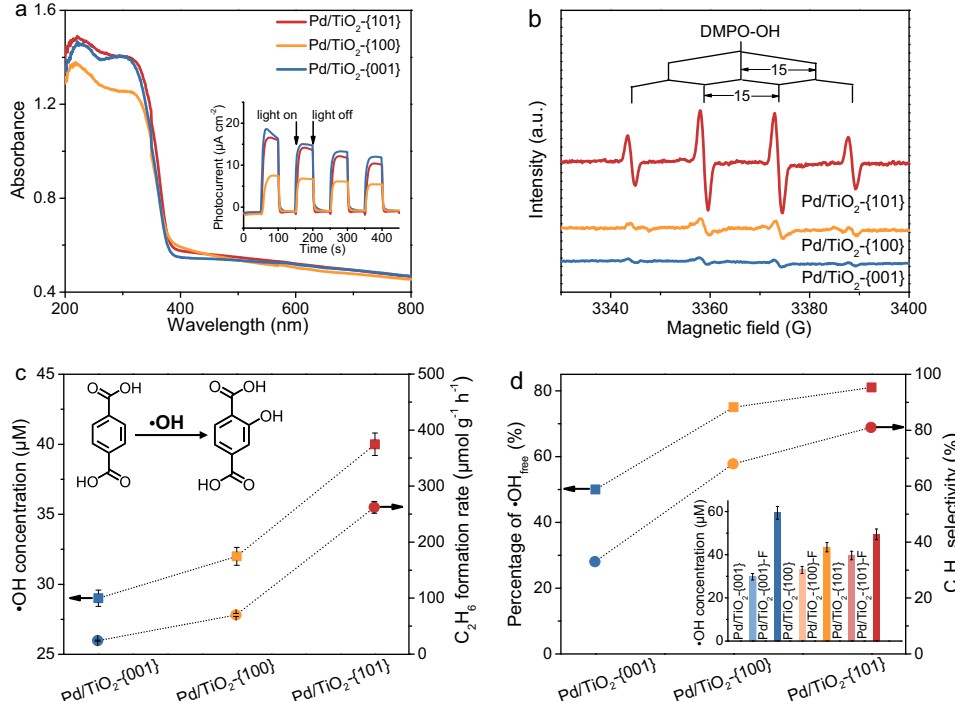

**Fig. 3 | Structure-property correlation. a** UV-Vis diffuse reflectance spectra and transient photocurrent responses (inset) for Pd/TiO$_2$ catalysts with different exposed anatase facets. **b** In situ ESR spectra using DMPO as a trapping agent under light irradiation for Pd/TiO$_2$ catalysts with different exposed anatase facets. **c** The relationship between C$_2$H$_6$ formation rate and concentration of •OH measured by the fluorescence method using terephthalic acid as a probe molecule. **d** The relationship between C$_2$H$_6$ selectivity and percentage of liquid-phase •OH. The inset of Fig. 3d displays the •OH concentrations in the absence and presence of fluoride ions. The error bar represents the relative deviation, which is within 5%.

on the surface into the solution[39,40]. The ratio of the •OH radicals existing in the liquid phase in all the •OH radicals generated could be evaluated and was found to depend on the exposed facet of TiO$_2$ (Fig. 3d). Such a ratio was the highest for the Pd/TiO$_2$-{101} catalyst. It is of interest that the ratio of the •OH in the liquid phase correlates well with the C$_2$ selectivity for the catalysts with different exposed facets of TiO$_2$ (Fig. 3d). These results enable us to propose that the •OH radicals existing in the liquid phase rather than adsorbed on catalyst surfaces are responsible for the formation of C$_2$ compounds.

Density functional theory (DFT) calculations were performed to gain in-depth insights into the reaction mechanism. We compared the adsorption strength of •OH radicals on three model surfaces of anatase, i.e., {001}, {100}, and {101} surfaces (Supplementary Fig. 16). Our calculations show that the adsorption energy of •OH radicals follows the trend of {001} (−1.57 eV) < {101} (−0.97 eV) < {100} (−0.72 eV) (Fig. 4a and Supplementary Fig. 17). Such strong adsorption indicates that the direct desorption of •OH from the anatase surface is energetically unfavorable. Alternatively, the •OH in the liquid phase may be generated by the reaction between the adsorbed •OH and a H$_2$O molecule near the surface via H transfer (inset of Fig. 4a). Our calculations show that such a H transfer reaction is exothermic and the reaction energies for the {001}, {100}, and {101} surfaces are −0.04, −0.16, and −0.22 eV, respectively (Fig. 4a and Supplementary Fig. 17). This result indicates that the liquid-phase •OH radicals would be formed more facilely in the case of TiO$_2$-{101} surface as compared to the other two surfaces and agrees well with the experimental fact that the fraction of liquid-phase •OH radicals is the highest for the Pd/TiO$_2$-{101} catalyst (Fig. 3d).

Our DFT calculations further reveal that the •OH radicals either adsorbed on the TiO$_2$-{101} surface or existing in liquid phase are capable of activating CH$_4$ via H abstraction to generate •CH$_3$ by overcoming barriers of 0.14 and 0.02 eV, respectively (Supplementary Fig. 18). On the other hand, the barrier for the cleavage of the C−H bond in CH$_4$ on

the Pd$_4$O cluster over a model Pd$_4$O/TiO$_2$-{101} catalyst is 0.53 eV (Supplementary Fig. 18). Thus, the •OH radical would be more favorable for initiating the C−H bond cleavage in our system. The selectivity of the whole reaction is determined by the fate of •CH$_3$. Considering the low concentration of •CH$_3$ radicals in the liquid phase, the probability of self-coupling of two liquid-phase •CH$_3$ radicals would be very limited. The encounter of •CH$_3$ radicals with other •OH radicals in liquid phase will cause the formation of a C−O bond rather than C−C bond, eventually leading to CO$_2$. Our DFT calculations reveal that the reaction between •CH$_3$ and H$_2$O, which leads to the formation of CH$_3$OH, is a thermodynamically unfavorable reaction ($\Delta G$ = +1.13 eV). A couple of studies have proposed that the metal co-catalyst may adsorb •CH$_3$ radicals and catalyze their subsequent transformations[12,20,41]. To further understand the potential role of Pd co-catalysts in the coupling of •CH$_3$ radicals, we performed DFT calculations using four models of Pd co-catalysts. The result shows that the isolated Pd$_1$ has the ability to trap only one •CH$_3$ radical (Fig. 4b), unfavorable for the coupling that needs two adjacent •CH$_3$ radicals. On the other hand, the DFT calculations demonstrate that adjacent •CH$_3$ radicals could be enriched on Pd(111) surface, supported Pd$_4$ cluster, and supported Pd$_4$O cluster with Pd−Pd pair sites from the viewpoint of thermodynamics (Fig. 4b). The energy barriers for the formation of C$_2$H$_6$ via coupling of the •CH$_3$ radicals adsorbed on Pd(111) surface, Pd$_4$ cluster, and Pd$_4$O cluster are calculated to be 1.93, 0.99, and 0.48 eV, respectively (Fig. 4b and Supplementary Fig. 19). Therefore, both the cluster feature of Pd co-catalysts and the presence of positively charged Pd$^{\delta+}$ sites play crucial roles in C$_2$ formation by affecting the enrichment and the coupling of •CH$_3$ radicals. Experimentally, when a physical mixture of the TiO$_2$-{101} sample and a Pd/Al$_2$O$_3$ catalyst, which itself was inactive for photocatalytic NOCM, was used instead of the Pd/TiO$_2$-{101} catalyst, CH$_4$ could be converted into C$_2$H$_6$ with 53% selectivity (Supplementary Table 4). This supports the result obtained from DFT calculations that the Pd co-catalyst with Pd$^{2+}$ contributes to aiding the coupling by trapping •CH$_3$ radicals from the liquid phase.

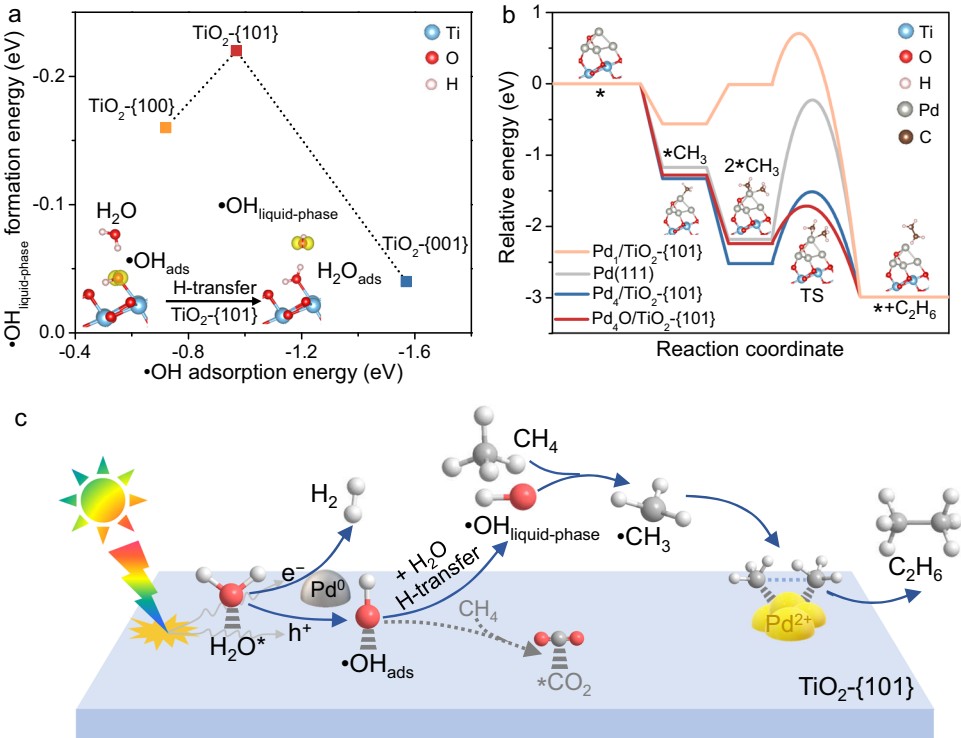

**Fig. 4 | DFT calculation results and reaction mechanism. a** The liquid-phase •OH formation energy versus the •OH adsorption energy on anatase $TiO_2$ with different exposed facets. The inset of Fig. 4a displays the optimized structures for •OH$_{liquid-phase}$ generation by H-transfer on $TiO_2$-{101} surface. **b** The relative energy for the coupling of •$CH_3$ to $C_2H_6$ on the model surfaces of $Pd_1$, $Pd_4$, and $Pd_4O$ loaded on anatase $TiO_2$-{101} as well as on the Pd(111) surface. The inset of Fig. 4b displays the optimized structures for •$CH_3$ enrichment and self-coupling on $Pd_4O$ site loaded on $TiO_2$-{101}. **c** Mechanism for photocatalytic NOCM with $H_2O$ on Pd/$TiO_2$-{101}.

Based on the above experimental and the DFT calculation results, we propose that the photocatalytic NOCM with $H_2O$ proceeds via a heterogeneous-homogeneous mechanism over the Pd/$TiO_2$-{101} catalyst (Fig. 4c). The generation of •OH radicals by the interaction of $H_2O$ with photogenerated holes has been observed. Our studies reveal that a part of •OH radicals are present in the liquid phase, and these •OH radicals are probably formed by the H transfer between the •OH radicals formed on semiconductor surfaces and the liquid-phase $H_2O$ near the surface. The structure-property correlation suggests that the •OH radicals in the liquid phase participate in the activation of $CH_4$ via H abstraction to form •$CH_3$. The Pd nanoparticles may function for catching •$CH_3$ radicals from the liquid phase and the presence of $Pd^{2+}$ sites on the nanoparticles favors the coupling of •$CH_3$ radicals by lowering the energy barrier. It is noteworthy that the Pd/$TiO_2$-{101} catalyst catalyzes the generation of liquid-phase •OH radicals more facilely as compared to other catalysts with different exposed facets, leading to the remarkable facet effect for the formation of $C_2$ compounds.

## Discussion

This work presents an efficient photocatalytic system for NOCM in the presence of $H_2O$, achieving a $CH_4$ conversion rate of 326 µmol g$^{-1}$ h$^{-1}$ at 81% $C_2$ selectivity, which represents one of the best values reported to date. We discovered a unique facet effect that the anatase $TiO_2$ nanocrystal mainly exposing {101} facet, which is usually considered as less active in photocatalysis, shows a significantly higher formation rate of $C_2$ compounds, whereas that mainly exposing high-energy {001} facet is rather less active and selective. The formation energy of •OH radicals in the liquid phase and the surface fraction of $Pd^{2+}$ are two key descriptors in the present system. It is revealed that the exposed facet can control the concentration of •OH radicals existing in the liquid phase during photocatalysis, and the anatase {101} facet offers a

significantly higher concentration of liquid-phase •OH radicals than the corresponding {001} and {100} facets. The liquid-phase •OH radical is proposed to be formed by the H transfer between the •OH radical adsorbed on the $TiO_2$ surface and a $H_2O$ molecule near the surface. A higher ratio of •OH radicals in the liquid phase to those on $TiO_2$ surfaces leads to higher $C_2$ selectivity, suggesting that the •OH radicals in the liquid phase account for the selective conversion of $CH_4$ to $C_2$ compounds, whereas those adsorbed on $TiO_2$ surfaces may contribute to the $CO_2$ formation. The Pd co-catalyst not only accelerates the electron-hole separation but also plays a role in enhancing the coupling of •$CH_3$ radicals by adsorbing and enriching the •$CH_3$ radicals from the liquid phase. The presence of positively charged $Pd^{2+}$ sites on the Pd nanoparticles is beneficial to lowering the energy barrier for the coupling of •$CH_3$ radicals accumulated on Pd nanoparticles.

## Methods
### Synthesis of catalysts
$TiO_2$-{101} nanocrystals were synthesized by a hydrothermal method. In brief, titanium isopropoxide (20 mL) and ultra-pure water (2.0 mL) were added into 100 mL Teflon-lined autoclave and mixed thoroughly by stirring, followed by hydrothermal treatment at 200 °C for 24 h. After the hydrothermal treatment, a white solid product was collected by centrifugation and repeated washing with water and ethanol. The $TiO_2$-{101} was obtained by calcining the solid product in air at 500 °C for 3 h.

$TiO_2$-{100} nanocrystals were synthesized by the following two-step method. In the first step, $TiCl_4$ (6.6 mL) was added into 20 mL HCl aqueous solution (0.43 M) at 0 °C. After stirring for 30 min, an $NH_3$ aqueous solution (5.5 wt%) was added, and then the pH value was adjusted to around 6.5 using an $NH_3$ aqueous solution (4.0 wt%). A white suspension was obtained and stirred for 2 h, followed by repeatedly washing until chloride ions were completely removed. A

solid Ti(OH)$_4$ precursor was collected after drying at 70 °C for 12 h. In the second step, Ti(OH)$_4$ (2.0 g) was dispersed into a mixed solution of (NH$_4$)$_2$SO$_4$ (0.50 g), water (15 mL), and isopropanol (15 mL). After being stirred for 1 h, the mixture was transferred into an autoclave (100 mL) and was subjected to hydrothermal treatment at 180 °C for 24 h. After hydrothermal synthesis, the solid product was collected by centrifugation, followed by repeated washing with water and ethanol and drying at 60 °C.

TiO$_2$-{001} nanocrystals were synthesized by a procedure reported previously[30]. Hydrofluoric acid (40 wt%) was employed as a capping agent. Tetrabutyl titanate (25 mL) and HF (3.0 mL) were mixed thoroughly to form a homogeneous solution and were subjected to hydrothermal treatment in an autoclave at 200 °C for 24 h. After hydrothermal synthesis, the solid product was collected by centrifugation, followed by repeated washing with water and ethanol. The sample was further dispersed in an aqueous solution of NaOH (0.1 M), followed by stirring for 24 h. The solid powders were recovered by centrifugation and washed with water to neutral, followed by drying at 60 °C.

The Pd/TiO$_2$ catalysts were prepared by an adsorption-reduction method. Typically, a PdCl$_2$ aqueous solution (concentration, 1.25 mg mL$^{-1}$; volume, 1.14 mL) was added dropwise into a mixture of TiO$_2$ nanocrystals synthesized above (100 mg) and water (30 mL). The pale-yellow solid with PdCl$_2$ adsorbed on TiO$_2$ (PdCl$_2$/TiO$_2$) was obtained after removal of H$_2$O. The obtained powdery precursor was calcined in air at 300 °C for 2 h and was reduced in H$_2$ flow at 300 °C for 2 h. The TiO$_2$ catalysts loaded with other metal co-catalysts, including Pt, Au, Ag, Cu, and Fe, were prepared by the same procedures except for using H$_2$PtCl$_6$, HAuCl$_4$, AgNO$_3$, Cu(NO$_3$)$_2$, and Fe(NO$_3$)$_3$ as the metal precursors instead of PdCl$_2$.

To investigate the effect of the fraction of Pd$^{2+}$ or the ratio of Pd$^{2+}$/Pd$^0$, we also prepared Pd-air/TiO$_2$-{101} and Pd-NaBH$_4$/TiO$_2$-{101} besides the standard Pd/TiO$_2$-{101} catalyst prepared by the H$_2$ reduction of the calcined PdCl$_2$/TiO$_2$. The Pd-air/TiO$_2$-{101} was obtained by calcining the PdCl$_2$/TiO$_2$ precursor at 300 °C for 2 h. The Pd-NaBH$_4$/TiO$_2$-{101} catalyst was obtained by reducing the PdCl$_2$/TiO$_2$ precursor with an aqueous solution of sodium borohydride (0.1 M).

### Photocatalytic reaction

Prior to each photocatalytic reaction, the catalyst (20 mg) was ultrasonically dispersed in ultra-pure water (50 mL) in a batch-type reactor (95 mL). The system was first evacuated and then purged with CH$_4$ (99.999%) for 10 min at a constant pressure. The reaction was performed by irradiation under a 300 W Xe lamp ($\lambda$ = 320-780 nm) for 4 h at room temperature. After the reaction, potential gaseous products (including C$_2$H$_6$, C$_2$H$_4$, C$_3$H$_8$, C$_3$H$_6$, CO$_2$, O$_2$, and H$_2$) were detected using a high-speed micro gas chromatograph (INFICON Micro GC Fusion) equipped with molecular sieve 5 A and Q-bond columns as well as a high-sensitivity thermal conductivity detector. Potential liquid organic products (including CH$_3$OH, HCHO, HCOOH) were quantitatively analyzed by $^1$H nuclear magnetic resonance (NMR) spectroscopy (Advance III 500-MHz Unity plus spectrometer, Bruker). Our analysis showed that the major products were C$_2$H$_6$, C$_2$H$_4$, H$_2$, and CO$_2$. No C$_3$ gaseous products or liquid organic compounds were observed in our photocatalytic system. The formation rates of the carbon-containing products were calculated on a carbon basis.

The performance parameters were calculated according to the following equations:

$$\text{CH}_4 \text{ conversion rate} = \left[2 \times n(\text{C}_2\text{H}_6) + 2 \times n(\text{C}_2\text{H}_4) + n(\text{CO}_2)\right]/m/t \tag{1}$$

$$\text{CH}_4 \text{ conversion} = \left[2 \times n(\text{C}_2\text{H}_6) + 2 \times n(\text{C}_2\text{H}_4) + n(\text{CO}_2)\right] \\ /n(\text{CH}_4) \times 100\% \tag{2}$$

$$\text{C}_2 \text{ selectivity} = \left[2 \times n(\text{C}_2\text{H}_6) + 2 \times n(\text{C}_2\text{H}_4)\right] \\ /\left[2 \times n(\text{C}_2\text{H}_6) + 2 \times n(\text{C}_2\text{H}_4) + n(\text{CO}_2)\right] \times 100\% \tag{3}$$

where $n$, $m$, and $t$ represent the amount of substance, mass of the catalyst, and reaction time.

The ratio of electrons and holes consumed in the reaction process was calculated using the following equation.

$$\frac{\text{e}^-}{\text{h}^+} = 2 \times n(\text{H}_2) / \left[2 \times n(\text{C}_2\text{H}_6) + 4 \times n(\text{C}_2\text{H}_4) + 8 \times n(\text{CO}_2)\right] \tag{4}$$

We measured the apparent quantum yield (AQY) of the Pd/TiO$_2$-{101} catalyst by using LED light for the photocatalytic NOCM to C$_2$H$_6$ and C$_2$H$_4$. The AQY for the formation of C$_2$H$_6$ and C$_2$H$_4$ was calculated using the following equation:

$$\text{AQY} = \left[2n(\text{C}_2\text{H}_6) + 4n(\text{C}_2\text{H}_4)\right] \times N_\text{A} / \left[I\left(\text{W cm}^{-2}\right) \\ \times S(\text{cm}^2) \times t(s)/E_\lambda(\text{J})\right] \times 100\% \tag{5}$$

where $n$, $N_\text{A}$, $I$, $S$, and $t$ represent the molar amounts of C$_2$H$_6$ and C$_2$H$_4$, Avogadro's constant, light intensity (0.20 W cm$^{-2}$), irradiation area (0.78 cm$^2$), and reaction time (3600 s), respectively. $E_\lambda$ is calculated using $hc/\lambda$ ($\lambda$ = 365 nm).

### Characterization

Powder X-ray diffraction (XRD) patterns were recorded on a Rigaku Ultima IV diffractometer. Raman spectroscopy measurements were carried out on a Renishaw inVia Raman microscope. Transmission electron microscopy (TEM) and scanning transmission electron microscopy (STEM) measurements were performed on a Tecnai F20 electron microscope (Phillips Analytical). X-ray photoelectron spectroscopy (XPS) was conducted using a Qtac-100 instrument, and all samples were under the protection of N$_2$ before XPS measurements. The content of each element was measured with an inductively coupled plasma optical emission spectrometer (ICP-OES, SPECTROBLUE FMX36). UV-Vis diffuse reflectance spectroscopy was measured on a Varian Cary 5000 spectrophotometer. Electron spin resonance (ESR) spectroscopic measurements were performed on a Bruker EMX-10/12 ESR spectrometer at room temperature. Photoluminescence spectra were recorded on a fluorescence spectrophotometer (F4500). N$_2$ physisorption measurements were performed on a Micromeritics Tristar 3020 surface area analyzer.

Measuring liquid-phase •OH radicals: The Pd/TiO$_2$ catalyst (20 mg) was dispersed in a 50 mL aqueous solution containing terephthalic acid (5.0 × 10$^{-4}$ M) and NaOH (2.0×10$^{-3}$ M). After stirring for 30 min in the dark, the suspension was irradiated under a 300 W Xe lamp ($\lambda$ = 320-780 nm). A small portion of the suspension (1.0 mL) was taken out and filtered after 0, 10, 20, and 30 min of irradiation. The filtrate (0.5 mL) was mixed with water (2.5 mL), and the solution was measured on a fluorescence spectrophotometer (F4500) at an excitation wavelength of 315 nm. Both the excitation and emission slit widths were 2.5 nm, and the scan speed was 1200 nm min$^{-1}$.

Measuring total •OH radicals: The method adopted for measuring the concentration of total •OH radicals was the same as that for measuring the liquid-phase •OH, except for using Pd/TiO$_2$-F instead of the original Pd/TiO$_2$. The Pd/TiO$_2$-F was prepared by the following procedure. An HF aqueous solution (40 wt%, 50 μL) was added to a mixture composed of Pd/TiO$_2$ (50 mg) and water (20 mL) under stirring, and the suspension was further stirred at room temperature for 4 h. It is well known that the hydroxyl groups on the surface of TiO$_2$ can be completely replaced by fluoride ions at a pH of 3-4. Surface fluorination can promote the desorption of surface-bound •OH radicals into the liquid phase. Therefore, the •OH radicals generated using Pd/TiO$_2$-F represent the concentration of total •OH radicals.

The density of oxygen vacancies was measured quantitatively by electronic titration using thionine acetate as a titrant[27]. Typically, the Pd/TiO$_2$ catalyst (5.0 mg) was ultrasonically dispersed in ultra-pure

water (50 mL) in a batch-type reactor (95 mL). The system was first evacuated and then purged with $CH_4$ (99.999%) for 10 min at a constant pressure. The reaction was performed by irradiation under a 300 W Xe lamp ($\lambda$ = 320-780 nm) for 2 h at room temperature. After the light irradiation, an aqueous solution of thionine acetate (0.3 mM, 1 mL) was quickly added to the suspension. The resulting suspension was filtrated, and the UV-vis absorption of the filtrate at 602.5 nm was measured. By comparing with the standard curve of different concentrations of thionine acetate aqueous solutions, the corresponding amount of thionine acetate consumed can be obtained. The density of oxygen vacancies was evaluated according to the correspondence that one oxygen vacancy is accompanied by two $Ti^{3+}$, and the oxidation of 2 moles of $Ti^{3+}$ requires 1 mole of thionine acetate[27].

The transient photocurrent response measurements were performed on an electrochemical workstation (CHI760E) in a standard three-electrode system. The Pt electrode and Ag/AgCl electrode were used as the counter and reference electrode, respectively. The $Na_2SO_4$ solution (0.5 mol/L) was used as the electrolyte. The working electrode was prepared as follows: the catalyst (10 mg) was dispersed in a mixture solution of Nafion perfluorinated resin solution (10 μL) and ethanol (1 mL). 100 μL of dispersion was uniformly dropped on the FTO plate (1 cm × 2 cm) so that the coating area was 1 cm$^{-2}$. An LED lamp with a wavelength of 365 nm served as a light source. The applied potential was 0 V vs. the Ag/AgCl reference electrode.

## Computational methods

The first principle spin-polarized calculations were carried out by using Vienna ab initio simulation package 6.3 (VASP 6.3)[42,43]. The exchange-correlation functional was described by generalized-gradient approximation (GGA) in the version of Perdew–Burke–Ernzerhof (PBE)[44]. The projector augmented-wave method (PAW)[45] was used to describe the ion-electron interaction with a plane wave basis set using a cut-off energy of 450 eV. As displayed in Supplementary Fig. 16, the anatase $TiO_2$-{001}, $TiO_2$-{100}, and $TiO_2$-{101} surfaces were modeled using a six-layer slab with (3 × 3) surface supercell ($Ti_{54}O_{108}$), a seven-layer slab with (3 × 1) surface supercell ($Ti_{42}O_{84}$), and an eight-layer slab with (2 × 3) surface supercell ($Ti_{48}O_{96}$), respectively. A 15 Å thick vacuum layer along the Z direction was adopted to avoid the interaction between adjacent images. To simulate the loaded Pd species with different valence states, the most stable $Pd_4$ cluster and $Pd_4O$ cluster were deposited on the $TiO_2$-{101} surface. For comparison, a single-atom Pd catalyst, i.e., $Pd_1$/$TiO_2$-{101}, was also constructed by replacing one Ti atom with Pd atom and removing one adjacent two-coordinated oxygen atom. The Pd-{111} surface was modeled with a 4 × 4 supercell and 4 layers.

During the structural optimization, the upper two layers with adsorbates were allowed to be fully relaxed while the other layers were fixed. The Brillouin zone of slab model was sampled by a Monkhorst–Pack scheme[46] with a grid of 3 × 3 × 1 k points. The Grimme's dispersion correction (DFT-D3) was utilized to include the van der Waals interaction[47]. The conjugate gradient method was used to optimize the positions of the ions until the residual force on each ion was less than 0.02 eV Å$^{-1}$. The total energy had been calculated up to an accuracy of $10^{-4}$ eV.

Generally speaking, PBE has the tendency to delocalize unpaired electrons due to the inherent self-interaction. To alleviate this issue, we used the DFT + U method[48] where a Hubbard-type correction was applied on both Ti-d and O-p orbitals. It was previously reported that U(d) = 4.2 eV for Ti and U(p) = 6.3 eV for O could be used to describe the polaronic states of $TiO_2$[49]. The spin density difference was utilized to identify a hole that was localized at the OH group. To locate the transition states (TSs), the climbing-image nudged elastic band method (CI-NEB) was applied with the convergence criteria of force of 0.05 eV Å$^{-1}$ for each atom[50].

The adsorption energy ($\Delta G_{ads}$) was defined using the following equation:

$$\triangle G_{ads} = G(\text{adsorbate}/\text{surface}) - G(\text{adsorbate}) - G(\text{surface}) \quad (6)$$

where $G$(adsorbate/surface), $G$(adsorbate), and $G$(surface) the free energies of the combined system, the gas-phase adsorbates, and the clean surface, respectively.

The liquid-phase •OH formation energy was calculated using the following equation:

$$\triangle G_f(\text{liquid-phase} \bullet \text{OH}) = G(TiO_2 - H_2O \cdots OH)$$
$$- G(TiO_2 - OH \cdots H_2O) \quad (7)$$

To calculate the reaction between •$CH_3$ and $H_2O$, leading to the formation of $CH_3OH$, we adopted the advanced Gaussian-3(G3) computational method with Gaussian 16 code[51,52], which is known for its accuracy in estimating molecular energies. We conducted studies using the B3LYP functional combined with the 6-311 G(d) basis set to perform geometry optimization and frequency calculation. Single-point energy calculation was performed at G3(MP2) level. The relevant chemical equation is:

$$\bullet CH_3 + H_2O = CH_3OH + \bullet H \quad (8)$$

## Reporting summary

Further information on research design is available in the Nature Portfolio Reporting Summary linked to this article.

## Data availability

All data supporting this work are available in the manuscript. Source data are provided in this paper.

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

## Acknowledgements

This work was supported by the National Key Research and Development Program of the Ministry of Science and Technology (No. 2022YFA1504500), the National Natural Science Foundation of China (Nos. 22121001, 22022201, 22132004, 92145301, 92045303, and 2072116), and the Fundamental Research Funds for the Central Universities (No. 20720220008).

## Author contributions

S.X., G.F., and Y.W. conceived and designed the study. H.Z. performed most of the experiments and data analysis. P.S. conducted theoretical calculations and analyzed the data. X.F., Z.H., W.Z., Q.G., and Y.Z. performed characterizations of photocatalysts and photocatalytic processes and analyzed the data. Q.Z. and X.W. provided guidance for experimental design. S.X. guided the experimental work and co-wrote the paper. G.F. guided the computational work and co-wrote the paper. Y.W. supervised the project and co-wrote the paper. All of the authors discussed the results and reviewed the manuscript.

## Competing interests

The authors declare no competing interests.
