## [Peer Review File · Nature Communications]

Unusual facet and co-catalyst effects in TiO₂-based photocatalytic coupling of methaneREVIEWER COMMENTS

Reviewer #1 (Remarks to the Author):

Comments to the author:

The author's article investigates the use of palladium as a cocatalyst to expose different crystal planes and selectively form C2 compounds, addressing both inter-plane problems and related loading issues. This study is indeed intriguing. However, some concerns hinder me from recommending its immediate release. After a minor revision, I suggest accepting this paper. Below are the points that need to be addressed:

1. Has the author compared the photocatalytic efficiency of Pd-TiO₂ with crystal plane selectivity to the industry-standard P25 catalyst, known for its high efficiency?
2. How does the author control the content of Pd²⁺ loaded on the TiO₂ surface?
3. Are the differences in photocatalytic activities between TiO₂ samples with nanosheets, nanorods, and nanopyramid morphologies attributed to morphology or crystal surface? Please explain.
4. The article mentions ·OH in the liquid phase; is the entire process solely a free radical process? If not, what is the contribution rate of free radicals to activity? If it relies on free radicals, will the reaction stop after adding a ·OH capture agent?
5. What is the light utilization rate during the photocatalytic process?
6. Kindly provide an explanation of the meaning of all lines in Figure 2 for each figure.
7. Considering various methods to load precious metals on catalysts, why did the author opt for the adsorption-reduction method?
8. Correct the representation in Fig. 3c (Structure-property correlation) to meet standard requirements.
9. Please explain the reason behind the XPS spectrogram showing Pd mainly present in Pd²⁺ and Pd⁰ on the Pd-loaded TiO₂ catalyst.

Reviewer #2 (Remarks to the Author):

Herein, the authors have synthesized TiO₂ nanosheet, nanorod, and nanobipyramid morphologies with mainly exposed facets of {001}, {100} and {101}, respectively and used for photocatalytic coupling of methane to ethylene and ethane. The authors claimed that anatase TiO₂ nanobipyramid containing {101} facet showed highest activity compared to other facets. The photogenerated holes are responsible for hydroxyl radical generation from water whereas in the conduction band HER happened. The presence of Pd²⁺ in the co-catalyst facilitated the coupling of methyl radicals, formed from the reaction of hydroxyl radical present in the solution and methane. The author has further performed theoretical calculations to prove the mechanism. Although the work is substantial, the lack of rigorous logical deduction and attention to detail increases the comprehension barrier of the article. However, considering the work's innovative approach towards facet engineering and its great interest in the synthesis of high-value-added products, the work can be published in the Nature Communications after addressing the following questions.

1. If the hydroxyl radicals are responsible for the methyl radical formation from methane, is it possible to form methyl radicals using hydrogen peroxide? Hydroxyl radicals are known for oxygen formation. Is the production of oxygen observed during the reaction?
2. Methyl radicals are highly reactive and short-lived species. Why is it not reacting with

abundant water molecules in aqueous solvent to form methanol?

3. TEM is a local imaging technique. The d spacing from a selected area does not mean the highest exposure of the facet. Moreover, why are there no changes between the XRD spectra of these different morphologies of TiO₂? In all the cases the highest exposed plane is (101).
4. The photoexcitation process does not stop in presence of N₂. Then, why did the HER reaction decrease after N₂ purging?
5. What are the other products other than C₂ products?
6. The XPS fittings of Pd XPS are not good. There are considerable data points outside the fitted region. Is the Pd⁴⁺ also present in the sample?
7. Why are the photoexcited electrons not reducing the Pd²⁺ present in the co-catalyst?
8. The ratio of the e⁻/h⁺ was calculated on the basis that all the hydroxyl radicals react with methane molecules to produce methyl radical. However, there will be some surface adsorbed hydroxyl radicals as well as all the hydroxyl radicals in solution will not participate in methyl radical formation. The authors should consider this part.

Reviewer #3 (Remarks to the Author):

The direct CH₄ conversion to value-added C₂ products (ethane or ethylene) has become one of most attractive research goals. Interestingly, the photocatalytic non-oxidative coupling of CH₄ (NOCM) to C₂ products is highly appealing because it additionally produces H₂ without CO₂ emissions and operates under mild reaction conditions. In the present work, authors studied the role of {101}, {001} and {100} facets of TiO₂ for photocatalytic NOCM to C₂ products using Pd as cocatalyst. Here, they found that the less active {101} facet is more reactive towards NOCM under their experimental conditions. They found that high amount of ·OH radicals formed in the liquid phase are the major active species for the observed high conversion rate in case of {101} facet compared to other facets. Further, they claimed that the Pd⁰ promotes H₂ production and Pd²⁺ facilitates the adsorption of ·CH₃ radicals to further promote their coupling to produce C₂ products.

Improving the selectivity of C₂ products by changing the facets of TiO₂ photocatalyst, studied in this work, is really an interesting and potentially significant contribution to photocatalytic NOCM. The role of TiO₂ facets for ·OH radical generation in liquid phase and their role in CH₄ conversion and selectivity to C₂ products has been explained in detail by insitu ESR and PL measurements. Moreover, the authors very nicely explained the role of Pd oxidation state (0 or 2+) towards the formation of C₂ products and H₂ under their experimental conditions. However, I am not fully convinced by the present version of the manuscript due to the following issues which need to be addressed for consideration.

1. Authors claimed that they obtained highest CH₄ conversion rate of 326 μmol g⁻¹ h⁻¹ with 81% selectivity of C₂ products (formation rate of ~264 μmol g⁻¹ h⁻¹). However, the cited Ref 21 (Nat Commun 13, 2806 (2022)) reported Pd¹/TiO₂ photocatalyst for NOCM to C₂ products, where they achieved C₂H₆ production rate of 910 μmol g⁻¹ h⁻¹, higher than

current report. Authors should discuss this work in their introduction and explain how their current work is different to Ref 21. The data in supplementary table 1 also missing this Pd1/TiO2 catalyst. Authors should reconsider their statement from the line 77 to 80.

2. Authors claimed that $\cdot\text{OH}$ radicals formed in the liquid phase initiate the C–H bond cleavage to form $\cdot\text{CH}_3$ radical, which then adsorb on Pd²⁺ to form C2 products. It is well documented that CH₄ can adsorb on metal surfaces like Pd easily. Therefore, I am not convinced fully to this claim as there is a possibility that CH₄ can adsorb on Pd surface and the photogenerated holes can break C–H bond to form $\cdot\text{CH}_3$ radicals. Authors should check the possibility of direct C–H bond oxidation over Pd surface of Pd/TiO₂ under their experimental conditions by performing some additional experiments.

3. Did authors check NOCM by replacing water with other solvents (especially some dry solvents)?

Reviewer #4 (Remarks to the Author):

There are some interesting results but I found the issues with experimental design. If one examines the TEM images the authors try to approximate fairly nonhomogeneous TiO₂ crystals with idealized shapes to derive preference of particular facets. It is not the right approach, there are single crystals that can be available for studies like this. Secondly I see a limited novelty of this study. Many decades ago there were studies of photocatalytic activities on different single crystal surfaces. If one removes this part of paper novelty, then there are already published papers for CH₄ coupling on Pd/TiO₂ photocatalyst. While there are some potentially novel elements in this paper, I do not see urgency or need to publish this paper in this high impact journal.

Responses to Reviewers

Response to Reviewer 1

General Comments: The author's article investigates the use of palladium as a cocatalyst to expose different crystal planes and selectively form C₂ compounds, addressing both inter-plane problems and related loading issues. This study is indeed intriguing. However, some concerns hinder me from recommending its immediate release. After a minor revision, I suggest accepting this paper. Below are the points that need to be addressed:

Reply and actions taken: We appreciate the positive comments raised by Reviewer 1 on our manuscript. Our replies to the detailed comments and the corresponding revisions are described as follows.

Comment 1: Has the author compared the photocatalytic efficiency of Pd-TiO₂ with crystal plane selectivity to the industry-standard P25 catalyst, known for its high efficiency?

Reply and actions taken: We thank the reviewer for this constructive comment. We have prepared Pd-modified commercial P25 catalyst by the adsorption-reduction method and have compared the photocatalytic performances of our Pd/TiO₂-{101} with Pd/P25 for the non-oxidative coupling of methane (NOCM). The result has been displayed in the **Supplementary Table 4** in the revised Supplementary Information. The result shows that Pd/P25 has similar activity or CH₄ conversion to Pd/TiO₂-{101}, but its selectivity of C₂ hydrocarbons is significantly lower than that of Pd/TiO₂-{101}. The C₂ yield over Pd/P25 is thus lower than that over Pd/TiO₂-{101}.

Supplementary Table 4 | Photocatalytic NOCM performances

Catalyst	Product amount (μmol)				CH ₄ conversion ^a (%)	C ₂ selectivity ^b (%)	C ₂ yield ^c (%)
	C ₂ H ₆	C ₂ H ₄	CO ₂	H ₂			
Pd/TiO ₂ -{101}	10	0.16	4.8	42	1.3	81	1.1
Pd/TiO ₂ -{100}	2.8	0	2.5	13	0.40	69	0.27
Pd/TiO ₂ -{001}	1.0	0	4.2	6.4	0.30	31	0.094
Pd/P25	8.1	0.26	13	61	1.4	56	0.77

Reaction conditions: catalyst, 20 mg; water, 50 mL; CH₄, 45 mL (2009 μmol); light source, 300 W Xe lamp (λ = 320-780 nm); irradiation time, 4 h.

^a CH₄ conversion = $[2 \times n(\text{C}_2\text{H}_6) + 2 \times n(\text{C}_2\text{H}_4) + n(\text{CO}_2)] / n(\text{CH}_4)$

^b C₂ selectivity = $[2 \times n(\text{C}_2\text{H}_6) + 2 \times n(\text{C}_2\text{H}_4)] / [2 \times n(\text{C}_2\text{H}_6) + 2 \times n(\text{C}_2\text{H}_4) + n(\text{CO}_2)]$

^c C₂ yield = CH₄ conversion × C₂ selectivity

We have added the following sentence in the revised manuscript to describe the result: “Further, we have compared the photocatalytic NOCM performance of the Pd/TiO₂-{101} with that of a Pd-promoted commercial P25 catalyst, which is known to possess high efficiencies in photocatalysis. The C₂ selectivity and yield of our Pd/TiO₂-{101} catalyst are significantly higher than those of Pd/P25 (Supplementary Table 4)” (please see Page 8, Lines 11-15).

Comment 2: How does the author control the content of Pd²⁺ loaded on the TiO₂ surface?

Reply and actions taken: To control the content (fraction) of Pd²⁺ loaded onto the TiO₂ surface, we have adopted different methods to post-treat the PdCl₂/TiO₂ precursor. First, PdCl₂ was adsorbed onto TiO₂ in aqueous solution, and then the PdCl₂/TiO₂ precursor was dried and post-treated by three different methods, i.e., (1) calcination in air at 300 °C for 2 h (denoted as Pd-air/TiO₂-{101}), (2) calcination in air at 300 °C for 2 h followed by H₂ reduction at 300 °C for another 2 h (denoted as Pd/TiO₂-{101}, the typical sample used in the present work), and (3) NaBH₄ reduction of the dried PdCl₂/TiO₂ precursor (denoted as Pd-NaBH₄/TiO₂-{101}). The fractions of Pd²⁺ on each sample was evaluated by XPS, which was measured by transferring the sample to XPS chamber without exposure to air. The Pd²⁺ fractions estimated from XPS measurements were 61%, 49%, and 33% for Pd-air/TiO₂-{101}, Pd/TiO₂-{101}, and Pd-NaBH₄/TiO₂-{101}, respectively (please see Supplementary Fig. 8 and Fig. 2d).

The following paragraph in the Method section has been modified in the revised manuscript to make the purpose of the preparation of three types of Pd/TiO₂-{101} catalysts clearer: “To investigate the effect of the fraction of Pd²⁺ or the ratio of Pd²⁺/Pd⁰, we also prepared Pd-air/TiO₂-{101} and Pd-NaBH₄/TiO₂-{101} besides the standard Pd/TiO₂-{101} catalyst prepared by the H₂ reduction of the calcined PdCl₂/TiO₂. The Pd-air/TiO₂-{101} was obtained by calcining the PdCl₂/TiO₂ precursor at 300 °C for 2 h. The Pd-NaBH₄/TiO₂-{101} catalyst was obtained by reducing the PdCl₂/TiO₂ precursor with an aqueous solution of sodium borohydride (0.1 M)” (please see Page 17, Paragraph 3). Further, the following sentences have been modified in the revised manuscript to describe the control of the content of Pd²⁺ more clearly: “The H₂ reduction of the calcined PdCl₂/TiO₂-{101} precursor, offered the Pd/TiO₂-{101} catalyst, which is typically adopted in the present work and has a surface Pd²⁺/Pd⁰ ratio close to 1.0 (Supplementary Table 2). On the other hand, the heat treatment in air and the reduction with NaBH₄ of the PdCl₂/TiO₂-{101} precursor offered catalysts (denoted as Pd-air/TiO₂-{101} and Pd-NaBH₄/TiO₂-{101}) with larger and smaller surface Pd²⁺/Pd⁰ ratios, respectively (Supplementary Fig. 8). The fractions of Pd²⁺ on the surfaces of Pd-air/TiO₂-{101}, Pd/TiO₂-{101}, and Pd-NaBH₄/TiO₂-{101} were

evaluated to be 61%, 49%, and 33%, respectively by XPS (Supplementary Fig. 8)” (please see from Page 9-the last line to Page 10-Line 7).

Comment 3: Are the differences in photocatalytic activities between TiO₂ samples with nanosheets, nanorods, and nanopyramid morphologies attributed to morphology or crystal surface? Please explain.

Reply and actions taken: In general, the morphology of a metal oxide determines the exposed facets (*Chem. Soc. Rev.* **43**, 1543-1574 (2014); *Chem. Rev.* **114**, 9559-9612 (2014)), and it may also affect the light absorption, charge separation, and specific surface area of a semiconductor (*Angew. Chem. Int. Ed.* **60**, 6160-6169 (2021); *ACS Catal.* **6**, 1097-1108 (2016)). Our studies have provided evidence that the difference in photocatalytic NOCM performance of Pd-promoted TiO₂ samples with different morphologies is mainly attributed to the difference in the exposed facet. The major experimental facts are as follows.

First, we found that there are no significant differences in the light absorption among the three Pd/TiO₂ catalysts as shown by the UV-Vis diffuse reflectance spectra (**Fig. 3a**). Second, the comparisons of the transient photocurrent response results (**Inset of Fig. 3a**) and the photoluminescence spectra (**Supplementary Fig. 11**) for the three catalysts with different morphologies show that the charge-separation efficiencies of Pd/TiO₂-{101} and Pd/TiO₂-{001} are similar, while that of Pd/TiO₂-{100} is lower. Third, the surface areas of Pd/TiO₂-{101}, Pd/TiO₂-{100}, and Pd/TiO₂-{001} catalysts were measured to be 49, 94, and 65 m² g⁻¹, respectively (**Supplementary Table 6**). Our photocatalytic studies demonstrate that the Pd/TiO₂-{101} catalyst shows the best NOCM performance, followed by Pd/TiO₂-{100} and Pd/TiO₂-{001} (**Fig. 2a**). Therefore, the factors other than the light absorption, the electron-hole separation, and the surface area determine the photocatalytic NOCM activity. Further, the photocatalytic H₂ evolution was found to decrease in the sequence of Pd/TiO₂-{001}>Pd/TiO₂-{101}>Pd/TiO₂-{100}, different from the trend observed for photocatalytic NOCM reaction. The trend for photocatalytic H₂ evolution could be explained by the higher electron-hole separation abilities of Pd/TiO₂-{001} and Pd/TiO₂-{101} as well as the high surface energy of the TiO₂-(001) facet. We have calculated the specific C₂ formation rate on the basis of the catalyst surface area, and the values are 5.3, 0.74, and 0.37 μmol m⁻² h⁻¹ for Pd/TiO₂-{101}, Pd/TiO₂-{100}, and Pd/TiO₂-{001}, respectively (**Supplementary Table 6**), demonstrating the intrinsic superiority of the {101} facet of TiO₂ in the photocatalytic NOCM. Based on these experimental facts, we believe that the differences in photocatalytic activities between Pd-promoted TiO₂ samples with nanosheets, nanorods, and nanopyramid morphologies should be attributed to the differences in the exposed TiO₂ facets, which result in

different surface reactivities toward the NOCM reaction.

The results and discussion mentioned above were already displayed in the first paragraph of the section of “**Structure-property relationship and reaction mechanism**” in our manuscript (*please see from Page 10-Paragraph 2 to Page 11-Paragraph 1*). To make the point understood more clearly, we have modified the last sentence of this paragraph as follows: “*Therefore, it is not the shape-based parameters, which may affect the light absorption, the electron-hole separation, and the surface area, but the exposed facet-based factor (not just the surface energy) plays a determining role in the TiO₂-catalysed NOCM*” (*please see Page 11, Lines 7-10*).

Comment 4: The article mentions •OH in the liquid phase; is the entire process solely a free radical process? If not, what is the contribution rate of free radicals to activity? If it relies on free radicals, will the reaction stop after adding a •OH capture agent?

Reply and actions taken: We thank the reviewer for this pertinent comment. Following the comment raised by the reviewer, we have investigated the influence of •OH radical scavenger. It is known that terephthalic acid can readily react with •OH to form 2-hydroxyterephthalic acid and thus can be used as a suitable scavenger to capture •OH radicals (*Electrochem. Commun.* **2**, 207-210 (2000); *J. Hazard. Mater.* **150**, 62-67 (2008)). We found that after 2 h of irradiation on our original photocatalytic system containing the Pd/TiO₂-{101} catalyst, CH₄ gas, and pure water, the amounts of C₂H₆, C₂H₄, and CO₂ formed were 6.4, 0.16, and 3.1 μmol, respectively. When we replaced pure water with an aqueous solution of terephthalic acid (5.0×10⁻⁴ M) while keeping other conditions unchanged, the amounts of C₂H₆, C₂H₄, and CO₂ formed decreased remarkably to 0.12, 0, and 0.28 μmol, respectively. In other words, the photocatalytic methane conversion activity decreased drastically, when terephthalic acid, an •OH radical scavenger, was added into the reaction system, providing evidence that •OH radicals mainly account for the methane conversion in our present system. Further, our additional experiments using dimethyl sulfoxide (DMSO), *N,N*-dimethylformamide (DMF), or perfluorohexanes to replace water as the solvent showed no formation of C₂ products, suggesting the crucial role of water. Thus, we propose that the photo-generated holes mainly work for the oxidation of water to form •OH radicals responsible for CH₄ activation, rather than directly activating CH₄.

Based on the results and discussion described above, we have added the following sentences in the revised manuscript: “*The substitution of H₂O by dimethyl sulfoxide (DMSO), N,N-dimethylformamide (DMF), or perfluorohexanes as the solvent resulted in no formation of C₂ products, further pointing out the determining role of H₂O. When terephthalic acid (25 μmol), a scavenger of •OH radicals^{35,36}, was added to the photocatalytic system with the Pd/TiO₂-{101} catalyst, the amounts of C₂H₆, C₂H₄, and*

CO₂ formed after 2 h of irradiation decreased drastically from 6.4, 0.16, and 3.1 μmol to 0.12, 0, and 0.28 μmol, respectively. We propose that the •OH radical formed in the presence of H₂O may participate in the activation of CH₄ and the formation of C₂ compounds” (please see Page 11, Paragraph 2, Lines 5-12).

Comment 5: What is the light utilization rate during the photocatalytic process?

Reply and actions taken: We thank this reviewer for kindly reminding us of estimating the light utilization rate for the present photocatalytic system. We have performed experiments to evaluate the apparent quantum yield (AQY) of the Pd/TiO₂-{101} catalyst for photocatalytic NOCM to C₂H₆ and C₂H₄. The AQY value for the formation of C₂ compounds was evaluated to be 0.21% at a wavelength of 365 nm.

The experimental method adopted for measuring the AQE has been described in the “**Method**” section (sub-section: “**Photocatalytic reaction**”) in the revised manuscript (*please see Page 18, Paragraph 4*). We have also added the following sentence in the revised main text to describe the experimental result: “*We have measured the apparent quantum efficiency (AQY) of the Pd/TiO₂-{101} catalyst for the formation of C₂H₆ and C₂H₄, and the AQY value at a wavelength of 365 nm is 0.21%*” (*please see Page 8, Paragraph 1, the last sentence*).

Comment 6: Kindly provide an explanation of the meaning of all lines in Figure 2 for each figure.

Reply and actions taken: We appreciate this comment raised by the reviewer. We have added arrows and text notes in **Fig. 2** in the revised manuscript (*please see Page 27, Fig. 2*). Now, the meaning of all the lines in Fig. 2 can be clearly understood (*please also see the revised Fig. 2 displayed in the next page*).

Fig. 2 | Photocatalytic behaviours. **a**, Effect of exposed anatase facet on NOCM performances over Pd/TiO₂ catalysts. **b**, Time courses for CH₄ conversion and C₂ selectivity over Pd/TiO₂ catalysts with different exposed anatase facets. **c**, Effect of Pd content on NOCM performances over Pd/TiO₂-{101} catalysts. **d**, Effect of the percentages of Pd²⁺ on Pd nanoparticles on NOCM performances and transient photocurrent densities over Pd/TiO₂-{101} catalysts. Reaction conditions: catalyst, 0.020 g; water, 50 mL; CH₄, 45 mL (2009 μmol); light source, 300 W Xe lamp (λ = 320-780 nm). The error bar represents the relative deviation, which is within 5%.

Comment 7: Considering various methods to load precious metals on catalysts, why did the author opt for the adsorption-reduction method?

Reply and actions taken: As compared to the adsorption-reduction (AR) method adopted in our work, the photo-deposition (PD) is a more general method employed for preparing the noble metal-supported semiconductor photocatalysts or depositing noble metals onto semiconductors for photocatalysis (*Chem. Rev.* **116**, 14587-14619 (2016)). Actually, we had already adopted the PD method to prepare the Pd-modified TiO₂-{101} catalyst. The photocatalytic NOCM performance of the Pd(PD)/TiO₂-{101} catalyst, however, was found to be poorer than that of Pd(AR)/TiO₂-{101} (**Table R1**). Our characterizations showed that the distribution of sizes of Pd nanoparticles loaded on TiO₂ by the PD method was broad (3-9 nm) and the average Pd particle size was 6.0 nm (**Figure R1**). On the other hand, by the AR method adopted in the present work, the distribution of Pd particles on the surface of TiO₂-{101} was significantly narrower

(1.2-3.0 nm) and the average size of Pd nanoparticles was 2.2 nm (**Fig. 1d**), much smaller than that by the PD method. Moreover, we compared the changes of the surface Pd²⁺ fraction estimated by XPS measurements on the Pd(PD)/TiO₂-{101} and Pd(AR)/TiO₂-{101} catalysts before and after photocatalytic NOCM. The fractions of Pd²⁺ on the Pd(AR)/TiO₂-{101} catalyst before and after the reaction were 48.9% and 45.4%, respectively (**Figure R2** and **Table R2**), suggesting that Pd²⁺ is relatively stable during the photocatalytic reaction. On the other hand, the fraction of Pd²⁺ on the Pd(PD)/TiO₂-{101} catalyst declined significantly from 39.2% to 27.5% after the photocatalytic NOCM (**Figure R2** and **Table R2**). Thus, the Pd²⁺ site on the Pd(PD)/TiO₂-{101} catalyst is more easily photo-reduced to Pd⁰. Our studies have already demonstrated that a higher surface fraction of Pd²⁺ is crucial to obtaining a higher C₂ formation rate and higher C₂ selectivity (**Fig. 2d**). We believe that the larger Pd particles and the facile photo-reduction of Pd²⁺ to Pd⁰ during the reaction are two major reasons for the poorer performance of the Pd(PD)/TiO₂-{101} catalyst. Therefore, we have chosen the adsorption-reduction method to load Pd onto TiO₂-{101}. To keep the main story of the present manuscript clear, we have not included the results for the Pd(PD)/TiO₂-{101} catalyst in the manuscript.

Table R1. Photocatalytic NOCM performances of Pd(AR)/TiO₂-{101} and Pd(PD)/TiO₂-{101}.

Catalyst	Product amount (μmol)				CH ₄ conversion ^a (%)	C ₂ selectivity ^b (%)
	C ₂ H ₆	C ₂ H ₄	CO ₂	H ₂		
Pd(AR)/TiO ₂ -{101}	10	0.16	4.8	42	1.3	81
Pd(PD)/TiO ₂ -{101}	5.6	0	8.7	59	1.1	56

Reaction conditions: 20 mg catalyst; 50 mL water; 45 mL (2009 μmol) CH₄; light source, 300 W Xe lamp (λ = 320-780 nm), irradiation time, 4 h. AR: adsorption-reduction; PD: photo-deposition.

^a CH₄ conversion = [2 × n(C₂H₆) + 2 × n(C₂H₄) + n(CO₂)] / n(CH₄);

^b C₂ selectivity = [2 × n(C₂H₆) + 2 × n(C₂H₄)] / [2 × n(C₂H₆) + 2 × n(C₂H₄) + n(CO₂)].

Figure R1. Scanning transmission electron microscopy (STEM) image and Pd particle-size distribution for Pd(PD)/TiO₂-{101}.

Figure R2. Pd XPS spectra. **a**, Pd(AR)/TiO₂-{101}-fresh. **b**, Pd(AR)/TiO₂-{101}-used. **c**, Pd(PD)/TiO₂-{101}-fresh. **d**, Pd(PD)/TiO₂-{101}-used. AR: adsorption-reduction; PD: photo-deposition.

Table R2. Percentages of Pd⁰ and Pd²⁺ on Pd/TiO₂ catalysts measured by XPS.

Catalysts	Pd ⁰		Pd ²⁺	
	Binding energy (eV)	Fraction (%)	Binding energy (eV)	Fraction (%)
Pd(AR)/TiO ₂ -{101}-fresh	335.3	51.1	336.6	48.9
Pd(AR)/TiO ₂ -{101}-used	335.2	54.6	336.7	45.4
Pd(PD)/TiO ₂ -{101}-fresh	335.2	60.8	336.7	39.2
Pd(PD)/TiO ₂ -{101}-used	335.2	72.5	336.6	27.5

Comment 8: Correct the representation in Fig. 3c (Structure-property correlation) to meet standard requirements.

Reply and actions taken: We thank the reviewer for pointing out this spelling error. We have changed “Strcuture-property correlation” to “Structure-property correlation” in the revised Fig. 3 (*please see Page 28*).

Comment 9: Please explain the reason behind the XPS spectrogram showing Pd mainly present in Pd²⁺ and Pd⁰ on the Pd-loaded TiO₂ catalyst.

Reply and actions taken: The XPS measurements showed the co-existence of Pd²⁺ and Pd⁰ on surfaces of our Pd-loaded TiO₂ catalysts prepared by the adsorption-reduction method and the surface ratios of Pd²⁺/Pd⁰ were close to 1.0 (**Supplementary Table 2**). Our catalysts were prepared by the following procedures. PdCl₂ was first adsorbed on the surface of TiO₂ in aqueous solution, and then the PdCl₂/TiO₂ precursor was dried at 60 °C and calcined in air at 300 °C for 2 h, followed by H₂ reduction at 300 °C for 2 h to obtain Pd/TiO₂ catalysts. The scanning transmission electron microscopy (STEM) showed that the average sizes of Pd loaded on TiO₂-{101}, TiO₂-{100}, and TiO₂-{001} were 2.2-2.5 nm (**Fig. 1d** and **Supplementary Fig. 4a, 4b**). Because the Pd nanoparticles were loaded by the adsorption of Pd precursors onto TiO₂, a large proportion of Pd atoms in the small Pd nanoparticles may be located near the interface and thus have strong interactions with O atoms on the TiO₂ surface (*ACS Catal.* **8**, 7556-7565 (2018); *Angew. Chem. Int. Ed.* **56**, 15993-15997 (2017)). The strong interaction between Pd and TiO₂ through the Pd–O bond would keep a part of Pd in Pd²⁺ state even after H₂ reduction at 300 °C. A previous work has also observed a similar phenomenon that Pd(2 nm)/TiO₂ contains both Pd²⁺ and Pd⁰, even after the reduction up to 500 °C (*ACS Catal.* **9**, 3946-3958 (2019)).

Based on the discussion above, we have added the following sentence in the revised manuscript: “*The high fraction of Pd²⁺ on our Pd/TiO₂ catalysts and the keeping of Pd²⁺ during the photocatalytic reaction could be attributed to the strong interaction*”

between the small Pd nanoparticles or nanoclusters and TiO₂^{32,33} (please see from Page 6-Line 1 from bottom to Page 7-Line 2).

Response to Reviewer 2

General Comments: Herein, the authors have synthesized TiO₂ nanosheet, nanorod, and nanobipyramid morphologies with mainly exposed facets of {001}, {100} and {101}, respectively and used for photocatalytic coupling of methane to ethylene and ethane. The authors claimed that anatase TiO₂ nanobipyramid containing {101} facet showed highest activity compared to other facets. The photogenerated holes are responsible for hydroxyl radical generation from water whereas in the conduction band HER happened. The presence of Pd²⁺ in the co-catalyst facilitated the coupling of methyl radicals, formed from the reaction of hydroxyl radical present in the solution and methane. The author has further performed theoretical calculations to prove the mechanism. Although the work is substantial, the lack of rigorous logical deduction and attention to detail increases the comprehension barrier of the article. However, considering the work's innovative approach towards facet engineering and its great interest in the synthesis of high-value-added products, the work can be published in the Nature Communications after addressing the following questions.

Reply and actions taken: We appreciate the positive comments on our manuscript raised by this reviewer. Our replies to the comments raised by the reviewer and the corresponding revisions are described as follows.

Comment 1: If the hydroxyl radicals are responsible for the methyl radical formation from methane, is it possible to form methyl radicals using hydrogen peroxide? Hydroxyl radicals are known for oxygen formation. Is the production of oxygen observed during the reaction?

Reply and actions taken: First, we would like to emphasize that all the experimental facts in the present work (including spectroscopic results and our additional result in the presence of an •OH radical scavenger, *please see Page 11, Paragraph 2, Lines 8-12*) support the mechanism via •OH radicals. Second, following the comment raised by the reviewer, we have performed control experiments to check whether C₂ products could be formed using hydrogen peroxide. After 4 h of irradiation, our original photocatalytic NOCM system containing the Pd/TiO₂-{101} catalyst, CH₄ gas, and pure water offered C₂H₆, C₂H₄, CO₂, and H₂ with amounts of 10, 0.16, 4.8, and 42 μmol, respectively (**Supplementary Table 4**). We have measured the formation of O₂ in our system, and no oxygen has been observed during the reaction. When an aqueous

solution of hydrogen peroxide (2.0 mM) was used instead of pure water and other reaction conditions were kept unchanged, no product was formed under dark reaction conditions, while under irradiation the obtained product only contained CO₂ (52 μmol) and O₂ (37 μmol) (**Table R3**). These experimental results indicate that our original system in the presence of water but in the absence of oxygen is more favorable for the selective coupling of CH₄ to C₂ products. When H₂O₂ is introduced into the reaction system, H₂O₂ decomposes to produce oxygen under irradiation, mainly resulting in the over-oxidation of CH₄ under aerobic conditions. Therefore, although H₂O₂ may provide •OH under irradiation, the use of H₂O₂ is not a good choice for our system because of the unfavorable over-oxidation. In our system with pure water, no release of O₂ has been observed, indicating that the generated •OH radicals are used efficiently for CH₄ activation and conversion.

Table R3. Effect of addition of H₂O₂ on photocatalytic NOCM performances of the Pd/TiO₂-{101} catalyst.

Reaction solution	Light	Product amount (μmol)				
		C ₂ H ₆	C ₂ H ₄	CO ₂	O ₂	H ₂
Pure water	Yes	10	0.16	4.8	0	42
H ₂ O ₂ solution (2.0 mM)	No	0	0	0	0	0
H ₂ O ₂ solution (2.0 mM)	Yes	0	0	52	37	0

Reaction conditions: 20 mg Pd/TiO₂-{101} catalyst; 50 mL reaction solution; 45 mL (2009 μmol) CH₄; light source, 300 W Xe lamp (λ = 320-780 nm); reaction time, 4 h.

Comment 2: Methyl radicals are highly reactive and short-lived species. Why is it not reacting with abundant water molecules in aqueous solvent to form methanol?

Reply and actions taken: We thank the reviewer for this constructive comment. It is noteworthy that our photocatalytic non-oxidative coupling of methane (NOCM) reaction is carried out in the presence of water, but in the absence of oxygen. According to the results reported in literature, the photocatalytic selective conversion of methane to methanol can usually be achieved when oxygen and water are present together (*J. Am. Chem. Soc.* **141**, 20507-20515 (2019); *J. Am. Chem. Soc.* **144**, 740-750 (2022); *ACS Catal.* **10**, 14318-14326 (2020); *Green Chem.* **23**, 3526-3541 (2021)). It is generally believed that O₂ reacts with photo-generated electrons and protons from water molecules to generate mild reactive oxygen species, •OOH, which may contribute to the selectivity formation of CH₃OH.

In our system, we have not observed any formation of CH₃OH. We have performed

DFT calculations to investigate the possibility of the reaction between $\bullet\text{CH}_3$ radicals with water molecules to form methanol. The calculation details are as follows. We adopted the advanced Gaussian-3(G3) computational method, which is known for its accuracy in estimating molecular energies. We conducted our calculations using the B3LYP functional combined with the 6-311G(d) basis set to perform geometry optimization and frequency calculation. Then, we performed single point energy calculation at G3(MP2) level. Our results indicate that the reaction between $\bullet\text{CH}_3$ with H_2O , leading to the formation of either CH_3OH or CH_4 , is thermodynamically unfavorable. The relevant chemical equations and their associated Gibbs free energy changes are as follow.

Moreover, H_2O molecules tend to cluster together because of their polarity and the formation of hydrogen bonds, which could make it challenging for $\bullet\text{CH}_3$ radicals to come into contact and react with H_2O . This could explain why we have not observed any formation of CH_3OH even in the presence of abundant H_2O molecules in our system.

Based on the results and discussion described above, we have added the DFT calculation method for the reaction between $\bullet\text{CH}_3$ and H_2O in the section of “**Computational methods**” in the revised **Supplementary Information**. The following sentence has been added in revised Manuscript to briefly mention the DFT calculation result: “*Our DFT calculations reveal that the reaction between $\bullet\text{CH}_3$ and H_2O , which leads to the formation of CH_3OH , is a thermodynamically unfavourable reaction ($\Delta G = +1.13 \text{ eV}$)” (please see Page 13, Paragraph 2, Lines 10-12).*

Comment 3: TEM is a local imaging technique. The d spacing from a selected area does not mean the highest exposure of the facet. Moreover, why are there no changes between the XRD spectra of these different morphologies of TiO_2 ? In all the cases the highest exposed plane is (101).

Reply: It is known that the powder XRD is a technique for the characterization of bulk crystalline structure, and it can hardly provide direct information on the exposed facets of nanocrystals, unless the crystalline sample is thin enough to become a 2D nanocrystal and this is not case for the nanocrystals in our work. Actually, many other studies have also pointed out that it is difficult to distinguish the exposed facets of TiO_2 nanocrystals by XRD (please see for examples: *Angew. Chem. Int. Ed.* **60**, 6160-6169 (2021); *Chem* **6**, 3038-3053 (2020)). Instead, TEM and HRTEM are generally adopted for the characterizations of the morphology and the preferentially exposed facets of nanocrystals. Our TEM measurements clearly show that almost all of the TiO_2

nanocrystals (>90%) synthesized in the present work possess uniform morphologies of nanosheet, nanorod, and nanobipyramid (**Supplementary Fig. 1 is just a representative**). Moreover, clear crystalline fringes are observed over each sample in the HRTEM image (**Fig. 1a-1c**). Based on the lattice spacing, we could determine that the TiO₂ samples with nanosheet, nanorod, and nanobipyramid morphologies mainly expose facets of {001}, {100}, and {101}, respectively (**Fig. 1a-1c**). A detailed analysis of the TEM results indicates that the fraction of the mainly exposed facet exceeds 80% in each TiO₂ sample (**Supplementary Fig. 2**). Such an approach to determining the exposed facets of TiO₂ nanocrystals has also been adopted in the reported literature (please see for examples: *J. Mater. Chem. A* **1**, 10532-10537 (2013); *ChemSusChem* **7**, 618-626 (2014)).

Comment 4: The photoexcitation process does not stop in presence of N₂. Then, why did the HER reaction decrease after N₂ purging?

Reply and actions taken: After 4 h of irradiation, our original photocatalytic NOCM system containing the Pd/TiO₂-{101} catalyst, CH₄ gas, and pure water offered C₂H₆, C₂H₄, CO₂, and H₂ with amounts of 10, 0.16, 4.8, and 42 μmol, respectively (**Supplementary Table 4**). When N₂ was used instead of CH₄ while keeping other conditions unchanged, only a small amount of H₂ (2.8 μmol) and a trace amount of O₂ were formed (**Supplementary Table 5**). Our studies suggest that in the photocatalytic NOCM reaction in the presence of water, the photogenerated electrons are used for the reduction of H₂O to H₂, while the photogenerated holes are used for the oxidation of H₂O to generate •OH radicals for the activation and conversion of CH₄. The oxidative and reductive half-reactions of our photocatalytic system are well coupled and can proceed efficiently at the same time. On the other hand, when CH₄ is replaced by N₂, the photogenerated electrons and holes are consumed for the reduction and oxidation of H₂O to H₂ and O₂, respectively. We only detected a trace amount of O₂, suggesting that the oxidation of H₂O to O₂ by holes is quite difficult under N₂ atmosphere in the absence of CH₄. Therefore, as compared to the NOCM reaction, the consumption rate of photogenerated holes for the oxidation of H₂O to O₂ under N₂ atmosphere becomes significant lower. This also decreases the rate of H₂ evolution, because the recombination of photogenerated electrons and holes would be accelerated.

Based on the result and discussion described above, we have added the following sentence in the revised manuscript: “*We only detected a trace amount of O₂ in the absence of CH₄, suggesting that the oxidation of H₂O to O₂ by photogenerated holes proceeded much slower, and this also decelerated the half reaction of H₂ evolution by photogenerated electrons*” (*please see Page 7, Paragraph 2, Lines 12-14*).

Comment 5: What are the other products other than C₂ products?

Reply and actions taken: Our analysis using an INFICON Micro GC Fusion with thermal conductivity detector as well as molecular sieve 5 A and Q-bond columns can detect the gaseous products, including C₂H₆, C₂H₄, C₃H₈, C₃H₆, CO₂, O₂, and H₂, if they are formed. The ¹H nuclear magnetic resonance (NMR) analysis (Advance III 500-MHz Unity plus spectrometer, Bruker) can detect liquid products, including CH₃OH, HCHO, and HCOOH, if they are formed. Actually, we found that other than C₂ products (i.e., C₂H₆ and C₂H₄), only H₂ and CO₂ could be observed. We confirmed that no C₃ gaseous products or liquid oxygen-containing compounds were formed.

Based on the results and discussion described above, we have modified the descriptions for the product analysis. Please see the following sentences in the revised “**Method**” section in the revised manuscript: “*After the reaction, potential gaseous products (including C₂H₆, C₂H₄, C₃H₈, C₃H₆, CO₂, O₂, and H₂) were detected using a high-speed micro gas chromatograph (INFICON Micro GC Fusion) equipped with molecular sieve 5 A and Q-bond columns as well as a high-sensitivity thermal conductivity detector. Potential liquid organic products (including CH₃OH, HCHO, HCOOH) were quantitatively analyzed by ¹H nuclear magnetic resonance (NMR) spectroscopy (Advance III 500-MHz Unity plus spectrometer, Bruker). Our analysis showed that the major products were C₂H₆, C₂H₄, H₂, and CO₂. No C₃ gaseous products or liquid organic compounds were observed in our photocatalytic system*” (**please see Page 18, Lines 2-8**).

Comment 6: The XPS fittings of Pd XPS are not good. There are considerable data points outside the fitted region. Is the Pd⁴⁺ also present in the sample?

Reply and actions taken: We thank the reviewer for pointing out this problem. We have re-measured the Pd 3d XPS spectra for all the samples and re-conducted XPS fittings. All the samples after the preparation by the adsorption-reduction method were transferred to XPS chambers without exposure to air. As displayed in **Figure 1f** and **Supplementary Fig. 4c, 4d** in the revised manuscript, now the XPS fitting results for Pd 3d spectra become much better. The fractions of Pd²⁺ and Pd⁰ on the Pd/TiO₂ catalysts have also been re-evaluated based on the revised fitting results and the results are displayed in the **revised Supplementary Table 2**. The Pd²⁺/Pd⁰ ratios are close to 1.0, which are basically the same with the previous results, and thus there is no change in the main conclusion.

As regards the possible existence of Pd⁴⁺, we think that it is difficult to consider such a possibility. This is because our Pd/TiO₂ catalysts were prepared by the adsorption of PdCl₂ onto TiO₂, followed by calcination and finally H₂ reduction at 573 K. Considering that the Pd 3d_{5/2} binding energy for Pd⁴⁺ is at 338.6-339.3 eV (*J. Phys.*

Chem. C **125**, 20845-20854 (2021), we have tried to include the peak of Pd⁴⁺ in the fitting, but we could not find the existence of the peak at the position of 338.6-339.3 eV (**Fig. 1f** and **Supplementary Fig. 4c,d**). Therefore, our results do not support the presence of Pd⁴⁺ in our Pd/TiO₂ catalysts.

Comment 7: Why are the photoexcited electrons not reducing the Pd²⁺ present in the co-catalyst?

Reply and actions taken: We have measured the Pd 3d XPS spectra for the three Pd/TiO₂ catalysts after the photocatalytic NOCM reactions. It is noteworthy that all the samples after the reaction were transferred to the XPS chamber without exposure to air (in the glove box under the protection by N₂). The results are displayed in **Supplementary Fig. 5** and **Supplementary Table 3** in the revised Supplementary Information. The comparison of the fractions of Pd²⁺ on the Pd/TiO₂ catalysts before and after photocatalytic NOCM reactions showed that the Pd²⁺ fraction on the three Pd/TiO₂ catalysts did not undergo significant changes after photocatalytic NOCM reactions for 4 h (**Table R4**). Therefore, the experimental result reveals that Pd²⁺ on TiO₂ surfaces can be sustained during the photocatalytic NOCM reaction.

As regards the reason for why Pd²⁺ could be kept during the photocatalytic NOCM reaction, we believe that the strong interaction between smaller Pd particles and TiO₂ is the key. The STEM studies showed that the average sizes of Pd loaded onto the TiO₂-{101}, TiO₂-{100}, and TiO₂-{001} were 2.2-2.5 nm (**Fig. 1d** and **Supplementary Fig. 4a, 4b**). Because the Pd nanoparticles were loaded by the adsorption of Pd precursors onto TiO₂, a large proportion of Pd atoms in the small Pd nanoparticles may be located near the interface and thus have strong interactions with O atoms on the TiO₂ surface (*ACS Catal.* **8**, 7556-7565 (2018); *Angew. Chem. Int. Ed.* **56**, 15993-15997 (2017)). The strong interaction between Pd and TiO₂ through the Pd–O bond would keep a part of Pd in Pd²⁺ state even after H₂ reduction at 300 °C. A previous work has also observed a similar phenomenon that Pd(2 nm)/TiO₂ contains both Pd²⁺ and Pd⁰, even after the reduction up to 500 °C (*ACS Catal.* **9**, 3946-3958 (2019)).

Based on the results and discussion described above, we have added the following sentences in the revised manuscript: “*Furthermore, the fractions of Pd²⁺ on the Pd/TiO₂ catalysts basically kept unchanged after 4 h of photocatalytic NOCM reaction (Supplementary Fig. 5 and Supplementary Table 3). The high fraction of Pd²⁺ on our Pd/TiO₂ catalysts and the keeping of Pd²⁺ during the photocatalytic reaction could be attributed to the strong interaction between the small Pd nanoparticles or nanoclusters and TiO₂^{32,33}*” (please see from Page 6-Line 3 from bottom to Page 7-Line 2).

Table R4. Fractions of Pd⁰ and Pd²⁺ on the Pd/TiO₂ catalysts before and after photocatalytic NOCM reactions measured by XPS

Catalysts	Pd ⁰		Pd ²⁺	
	Binding energy (eV)	Fraction (%)	Binding energy (eV)	Fraction (%)
Pd/TiO ₂ -{101}-fresh	335.3	51.1	336.6	48.9
Pd/TiO ₂ -{101}-used	335.2	54.6	336.7	45.4
Pd/TiO ₂ -{100}-fresh	335.2	51.3	336.7	48.6
Pd/TiO ₂ -{100}-used	335.3	57.6	336.5	42.4
Pd/TiO ₂ -{001}-fresh	335.2	52.3	336.6	47.7
Pd/TiO ₂ -{001}-used	335.2	54.8	336.5	45.2

Reaction conditions: catalyst, 20 mg; water, 50 mL; CH₄, 45 mL (2009 μmol); light source, 300 W Xe lamp ($\lambda = 320\text{-}780$ nm); irradiation time, 4 h.

Comment 8: The ratio of the e⁻/h⁺ was calculated on the basis that all the hydroxyl radicals react with methane molecules to produce methyl radical. However, there will be some surface adsorbed hydroxyl radicals as well as all the hydroxyl radicals in solution will not participate in methyl radical formation. The authors should consider this part.

Reply and actions taken: We thank the reviewer for this pertinent comment. We have measured the possibly remaining •OH radicals after the photocatalytic reaction using the Pd/TiO₂-{101} catalyst. After 4 h of irradiation, we added 25 μmol of terephthalic acid into the reaction system to measure the amount of •OH radicals remaining in the system by the fluorescence method. The experimental result showed that there was no fluorescence signal peak corresponding to 2-hydroxyterephthalic acid at 423 nm (**Figure R3**), indicating that the residual •OH radicals in the system could be ignored. Further, when we calculate the ratio of e⁻/h⁺ by using the ratio of the number of electrons consumed in the formation of H₂ and the number of holes consumed in the carbon-containing products formed by CH₄ oxidation ($e^-/h^+ = 2 \times n(\text{H}_2) / [2 \times n(\text{C}_2\text{H}_6) + 4 \times n(\text{C}_2\text{H}_4) + 8 \times n(\text{CO}_2)]$), the number is close to 1.0, suggesting that this calculation method is reasonable.

Figure R3. The photoluminescence spectrum for the reaction solution after reaction. Reaction conditions: Pd/TiO₂-{101} catalyst, 20 mg; water, 50 mL; CH₄, 45 mL (2009 μmol); light source, 300 W Xe lamp ($\lambda = 320\text{-}780$ nm); irradiation time, 4 h.

Response to Reviewer 3

General Comments: The direct CH₄ conversion to value-added C₂ products (ethane or ethylene) has become one of most attractive research goals. Interestingly, the photocatalytic non-oxidative coupling of CH₄ (NOCM) to C₂ products is highly appealing because it additionally produces H₂ without CO₂ emissions and operates under mild reaction conditions. In the present work, authors studied the role of {101}, {001} and {100} facets of TiO₂ for photocatalytic NOCM to C₂ products using Pd as cocatalyst. Here, they found that the less active {101} facet is more reactive towards NOCM under their experimental conditions. They found that high amount of $\cdot\text{OH}$ radicals formed in the liquid phase are the major active species for the observed high conversion rate in case of {101} facet compared to other facets. Further, they claimed that the Pd⁰ promotes H₂ production and Pd²⁺ facilitates the adsorption of $\cdot\text{CH}_3$ radicals to further promote their coupling to produce C₂ products.

Improving the selectivity of C₂ products by changing the facets of TiO₂ photocatalyst, studied in this work, is really an interesting and potentially significant contribution to photocatalytic NOCM. The role of TiO₂ facets for $\bullet\text{OH}$ radical generation in liquid phase and their role in CH₄ conversion and selectivity to C₂ products has been explained in detail by in situ ESR and PL measurements. Moreover, the authors very nicely explained the role of Pd oxidation state (0 or 2+) towards the formation of C₂ products and H₂ under their experimental conditions. However, I am not fully convinced by the present version of the manuscript due to the following issues which need to be addressed for consideration.

Reply and actions taken: We appreciate the positive and constructive comments raised by Reviewer 3. Our replies to the comments raised by the reviewer and the corresponding revisions are described as follows.

Comment 1: Authors claimed that they obtained highest CH₄ conversion rate of 326 $\mu\text{mol g}^{-1} \text{h}^{-1}$ with 81% selectivity of C₂ products (formation rate of $\sim 264 \mu\text{mol g}^{-1} \text{h}^{-1}$). However, the cited Ref 21 (Nat Commun 13, 2806 (2022)) reported Pd₁/TiO₂ photocatalyst for NOCM to C₂ products, where they achieved C₂H₆ production rate of 910 $\mu\text{mol g}^{-1} \text{h}^{-1}$, higher than current report. Authors should discuss this work in their introduction and explain how their current work is different to Ref 21. The data in supplementary table 1 also missing this Pd₁/TiO₂ catalyst. Authors should reconsider their statement from the line 77 to 80.

Reply and actions taken: We thank this reviewer for reminding us of the excellent result achieved in a reference, which was already cited as Ref. 21 in our manuscript. Because the photocatalytic performance depends significantly on reaction conditions, while the reaction conditions (such as the catalyst amount, the reaction time and temperature, the area and intensity of light irradiation) are usually different in different reported articles, it is difficult to compare fairly the reaction performances reported in different papers. Therefore, we agree with the reviewer that it is unsuitable to use the word like “the highest”.

In order to make a better comparison, we have further compared the C₂ production rate with the unit of “ $\mu\text{mol h}^{-1}$ ”, which is also adopted in many references, in addition to the unit of “ $\mu\text{mol g}^{-1} \text{h}^{-1}$ ”. The results for different photocatalysts have been added in **Supplementary Table 1** in the revised Supplementary Information. The Pd₁/TiO₂ catalyst reported in Ref. 21 shows a C₂H₆ formation rate of 910 $\mu\text{mol g}^{-1} \text{h}^{-1}$, corresponding to 2.7 $\mu\text{mol h}^{-1}$. Our Pd/TiO₂-{101} catalyst achieved a C₂ formation rate of 266 $\mu\text{mol g}^{-1} \text{h}^{-1}$, corresponding to 5.2 $\mu\text{mol h}^{-1}$. Therefore, we can say that the C₂ formation rate of our Pd/TiO₂-{101} catalyst is at least comparable to that of the Pd₁/TiO₂ catalyst reported in Ref. 21.

We have added the formation rate of C₂H₆ for the Pd₁/TiO₂ catalyst in **Supplementary Table 1** in the revised Supplementary Information. The following sentence in the last paragraph of **Introduction**: “*We achieve a CH₄ conversion rate of 326 $\mu\text{mol g}^{-1} \text{h}^{-1}$ with 81% selectivity of C₂ compounds over {101}-dominated anatase TiO₂ nanocrystals modified by Pd²⁺-containing co-catalyst and the attained C₂ formation rate is significantly higher than those reported to date (Supplementary Table 1)*” **has been revised to** “*We achieve a CH₄ conversion rate of 326 $\mu\text{mol g}^{-1} \text{h}^{-1}$ with 81% selectivity of C₂ compounds over {101}-dominated anatase TiO₂ nanocrystals modified by Pd²⁺-containing co-catalyst and the attained C₂ formation rate represents*

one of the best values reported to date (Supplementary Table 1)” (please see Page 5, Lines 5-8). Similarly, the following sentence in **Conclusion**: “This work presents an efficient photocatalytic system for NOCM in the presence of H₂O, achieving a CH₄ conversion rate of 326 μmol g⁻¹ h⁻¹ at 81% C₂ selectivity, significantly higher than those reported to date under similar conditions” **has been revised to** “This work presents an efficient photocatalytic system for NOCM in the presence of H₂O, achieving a CH₄ conversion rate of 326 μmol g⁻¹ h⁻¹ at 81% C₂ selectivity, which represents one of the best values reported to date” (Please see Page 15, the first sentence of **Conclusion**).

Regarding the difference between our present work and Ref. 21 reported by Xiong and co-workers, we think that each work has its own focus and uniqueness. Ref. 21 fabricated a single-atoms Pd-modified TiO₂ catalyst (Pd₁/TiO₂) for the photocatalytic NOCM and focused on **the role of single Pd sites in the photocatalytic NOCM reaction**. It is found that the Pd–O₄ site on TiO₂ surfaces could promote the formation of C₂H₆ by accumulating the photogenerated holes for CH₄ activation and reducing the contribution of O sites to the valence band to suppress the over-oxidation of CH₄ and to improve the stability. On the other hand, our present work focuses on **the effects of the exposed facet of TiO₂ nanocrystals and the cationic Pd species on the photocatalytic NOCM reaction**. We discovered a very unique facet effect that the anatase TiO₂ nanocrystal mainly exposing {101} facet, which is usually regarded as less active in photocatalysis, shows significantly higher formation rate of C₂ compounds than the nanocrystals mainly exposing other facets, whereas that mainly exposing high-energy {001} facet is rather less active and less selective. Our studies propose that the anatase {101} facet favors the formation of •OH radicals in aqueous phase near the surface for the activation of CH₄ to •CH₃ radicals. Further, we found the crucial role of Pd²⁺ in the formation of C₂ compounds. We propose that Pd⁰ accelerated the electron-hole separation and is responsible for H₂ formation, whereas Pd²⁺ accounts for the adsorption and coupling of •CH₃ radicals and the formation of C₂ compounds.

Based on the discussion described above, we have briefly discussed the work of Ref. 21 in the **Introduction** in the revised manuscript. The following sentence: “Some noble metal co-catalysts such as Pt¹⁶, Ag¹⁸, Au¹⁹, and Pd^{20,21} have been exploited for the metal oxide-based photocatalytic NOCM, but the structure requirement and functioning mechanism of the noble metal co-catalysts remain ambiguous” **has been revised to**: “Some noble metal co-catalysts such as Pt¹⁶, Ag¹⁸, Au¹⁹, and Pd^{20,21} have been exploited for the metal oxide-based photocatalytic NOCM. For example, a single-atom Pd-modified TiO₂ catalyst (Pd₁/TiO₂) showed an outstanding activity among different catalysts (Supplementary Table 1), and the accumulated photogenerated holes on Pd sites was proposed for the activation and conversion of CH₄ to C₂H₆²¹” (please

see Page 4, Paragraph 2, Lines 3-7).

Supplementary Table 1 | Representative results for the photocatalytic NOCM reaction reported to date.

Catalysts	CH ₄ conversion rate (μmol g ⁻¹ h ⁻¹)	C ₂₊ formation rate (μmol g ⁻¹ h ⁻¹)	C ₂₊ formation rate (μmol h ⁻¹)	C ₂₊ selectivity (%)	Ref.
Zn ²⁺ -ZSM-5	9.8	C ₂ H ₆ , 3.0	C ₂ H ₆ , 3.0	99	S15
Ga ³⁺ -ETS-10	30	C ₂ H ₄ , 0.6; C ₂ H ₆ , 11	C ₂ H ₄ , 0.12; C ₂ H ₆ , 2.2	~100	S16
Au/ZnO	24	C ₂ H ₆ , 12	C ₂ H ₆ , 0.012	~100	S17
Pt/TiO ₂	138	C ₂ H ₆ , 51; C ₂ H ₄ , 2.2	C ₂ H ₆ , 3.9; C ₂ H ₄ , 0.17	62	S18
Pt-TiO ₂ -SiO ₂	3.5	C ₂ H ₆ , 1.6	C ₂ H ₆ , 0.32	90	S19
Ag-HPW-TiO ₂	55	C ₂ H ₆ , 23; C ₃ H ₈ , 1.2	C ₂ H ₆ , 2.3; C ₃ H ₈ , 0.12	90	S20
ZnO-AuPd _{2.7}	79	C ₂ H ₆ , 25; C ₂ H ₄ , 13	C ₂ H ₆ , 0.05; C ₂ H ₄ , 0.026	96	S21
Nb-TiO ₂ -SiO ₂	3.6	C ₂ H ₆ , 1.7; C ₃ H ₈ , 0.07	C ₂ H ₆ , 0.17; C ₃ H ₈ , 0.007	96	S22
		C ₂ H ₄ , 0.05; C ₂ H ₆ , 1.7; C ₃ H ₆ , 0.55;	C ₂ H ₄ , 0.002; C ₂ H ₆ , 0.085; C ₃ H ₆ , 0.028;		
0.2Pt@BT-O	41	C ₃ H ₈ , 8.6; C ₄ H ₈ , 0.05; C ₄ H ₁₀ , 1.7; > C ₄ , 0.15	C ₃ H ₈ , 0.43; C ₄ H ₈ , 0.002; C ₄ H ₁₀ , 0.085; > C ₄ , 0.0075	99	S23
Pd₁/TiO₂	968	C₂H₆, 910	C₂H₆, 2.7	94	S24
Pd/TiO ₂ -{101}	326	C ₂ H ₆ , 262; C ₂ H ₄ , 4.0	C ₂ H ₆ , 5.2; C ₂ H ₄ , 0.08	81	This work

Comment 2: Authors claimed that •OH radicals formed in the liquid phase initiate the C–H bond cleavage to form •CH₃ radical, which then adsorb on Pd²⁺ to form C₂ products. It is well documented that CH₄ can adsorb on metal surfaces like Pd easily. Therefore, I am not convinced fully to this claim as there is a possibility that CH₄ can adsorb on Pd surface and the photogenerated holes can break C–H bond to form •CH₃ radicals. Authors should check the possibility of direct C–H bond oxidation over Pd surface of Pd/TiO₂ under their experimental conditions by performing some additional experiments.

Reply and actions taken: We thank the reviewer for this pertinent comment. We fully

understand the point raised by the reviewer, and we also considered previously that the activation of CH₄ on Pd surfaces by the photogenerated holes might be the major path for the formation of C₂ compounds. However, our detailed experimental and computational studies support the idea that •OH radicals mainly account for the activation of CH₄ to •CH₃ radicals, forming C₂ compounds via the coupling of •CH₃ radicals.

The experimental facts that support our idea are as follows. First, we found that H₂O plays a very crucial role in C₂ formation. The C₂ formation over the Pd/TiO₂-{101} catalyst was very low in the absence of H₂O and the presence of H₂O accelerated the C₂ formation rate for ~29 times (**Supplementary Fig. 12**). Our additional experiments using DMSO, DMF or PHF (perfluorohexanes) as the solvent instead of H₂O did not show the formation of C₂ compounds. Second, our additional experiments demonstrated that the reaction was significantly suppressed by adding •OH radical scavenger into the system. In brief, when terephthalic acid, a good •OH radical scavenger (*Electrochem. Commun.* **2**, 207-210 (2000); *J. Hazard. Mater.* **150**, 62-67 (2008)), was added into the photocatalytic system with the Pd/TiO₂-{101} catalyst, the amounts of C₂H₆, C₂H₄, and CO₂ formed after 2 h of irradiation decreased from 6.4, 0.16, and 3.1 μmol to 0.12, 0, and 0.28 μmol, respectively. Third, we already showed that there is a good relationship between the C₂H₆ formation rate and the concentration of •OH radicals in liquid phase (Fig. 3c).

We have added the following sentences in the revised manuscript to describe the additional experimental results: “*The substitution of H₂O by dimethyl sulfoxide (DMSO), N,N-dimethylformamide (DMF), or perfluorohexanes (PFH) as the solvent resulted in no formation of C₂ products, further pointing out the determining role of H₂O. When terephthalic acid (25 μmol), a scavenger of •OH radicals^{35,36}, was added to the photocatalytic system with the Pd/TiO₂-{101} catalyst, the amounts of C₂H₆, C₂H₄, and CO₂ formed after 2 h of irradiation decreased drastically from 6.4, 0.16, and 3.1 μmol to 0.12, 0, and 0.28 μmol, respectively. We propose that the •OH radical formed in the presence of H₂O may participate in the activation of CH₄ and the formation of C₂ compounds*” (**please see Page 11, Paragraph 2, Lines 5-12**).

Further, our computational results confirm that the barrier for the cleavage of C–H bond of CH₄ to form •CH₃ by •OH radicals is much lower than that for the cleavage of C–H bond of CH₄ on Pd surfaces of Pd/TiO₂. In brief, our DFT calculations utilizing the Vienna Ab Initio Simulation Package (VASP) code at the GGA/PBE level indicate that the adsorption of CH₄ on Pd surfaces is relatively weak, with an energy of only about –0.25 eV, while the barrier for the cleavage of C–H bond in CH₄ on Pd surfaces is approximately 0.75 eV. These results are basically consistent with those reported in literature (*Science* **373**, 1518-1523 (2021); *Nat. Catal.* **4**, 830-839 (2021); *Nat. Commun.*

14, 6343 (2023)). When the Pd₄O/TiO₂-{101} is used as the model catalyst, which is more pertinent to our system, the barrier for the cleavage of C–H bond in CH₄ on the Pd₄O cluster is 0.53 eV. In contrast, the barriers for the cleavage of C–H bond in CH₄ by the •OH radical adsorbed on the TiO₂ surface and the •OH radical in the liquid phase are calculated to be only 0.14 and 0.02 eV, respectively, both of which are significantly lower than those on the Pd surface or the Pd₄O cluster. Therefore, our DFT calculations demonstrate that the activation of CH₄ by •OH radicals is more favorable than that on Pd surfaces.

Based on the results and discussion described above, we have modified and added the following sentences in the revised manuscript: “Our DFT calculations further reveal that the •OH radicals either adsorbed on the TiO₂-{101} surface or existing in liquid phase are capable of activating CH₄ via H abstraction to generate •CH₃ by overcoming barriers of 0.14 and 0.02 eV, respectively (Supplementary Fig. 16). On the other hand, the barrier for the cleavage of the C–H bond in CH₄ on the Pd₄O cluster over a model Pd₄O/TiO₂-{101} catalyst is 0.53 eV (Supplementary Fig. 16). Thus, the •OH radical would be more favorable for initiating the C–H bond cleavage in our system” (please see Page 13, Paragraph 2, Lines 1-6).

Supplementary Fig. 16 | DFT calculations for the cleavage of the C–H bond in CH₄. **a**, The relative energies for the cleavage of the C–H bond in CH₄ to •CH₃ on Pd₄O cluster over TiO₂-{101} surface, by •OH adsorbed on the TiO₂-{101} surface, and by •OH radicals in the liquid phase. **b**, Structures of the initial state (IS), transition state (TS), and final state (FS) in the CH₄ activation process. Ti, O, H, Pd, and C are labelled as light blue, red, white, grey, and brown spheres, respectively.

Comment 3: Did authors check NOCM by replacing water with other solvents (especially some dry solvents)?

Reply and actions taken: We thanks the reviewer for this constructive comment.

Following the comment raised by the reviewer, we have employed some anhydrous solvents to replace H₂O to carry out the photocatalytic NOCM reaction with the Pd/TiO₂-{101} catalyst. The experimental result has been presented in **Table R5**. After 4 h of irradiation, the amounts of C₂H₆, C₂H₄, CO₂ and H₂ formed in the presence of H₂O were 10, 0.16, 4.8 and 42 μmol, respectively. When H₂O was replaced by dimethyl sulfoxide (DMSO), *N,N*-dimethylformamide (DMF), or perfluorohexanes, no C₂ compounds were formed. Significant amounts of CO and CO₂ were observed in the case of using DMF, which likely arose from the photocatalytic conversion of the solvent itself instead of CH₄.

Based on the result described above, we have added the following sentence in the manuscript to describe the experimental results: “*The substitution of H₂O by dimethyl sulfoxide (DMSO), N,N-dimethylformamide (DMF), or perfluorohexanes as the solvent resulted in no formation of C₂ products, further pointing out the determining role of H₂O*” (please see Page 11, Paragraph 2, Lines 5-8).

Table R5. Photocatalytic NOCM performances of Pd/TiO₂-{101} catalyst dispersed in different solvents.

Solvent	Product amount (μmol)				
	C ₂ H ₆	C ₂ H ₄	CO	CO ₂	H ₂
Water (H ₂ O)	10	0.16	0	4.8	42
Dimethyl sulfoxide (CH ₃ SOCH ₃)	0	0	0	1.1	2.5
N,N -Dimethylformamide (C ₃ H ₇ NO)	0	0	40	2.6	35
Perfluorohexanes (C ₆ F ₁₄)	0	0	0	1.9	9.0

Reaction conditions: 20 mg Pd/TiO₂-{101}; 50 mL solvent; 45 mL (2009 μmol) CH₄; light source, 300 W Xe lamp (λ = 320-780 nm), irradiation time, 4 h.

Response to Reviewer 4

General Comments: There are some interesting results but I found the issues with experimental design. If one examines the TEM images the authors try to approximate fairly nonhomogeneous TiO₂ crystals with idealized shapes to derive preference of particular facets. It is not the right approach, there are single crystals that can be available for studies like this. Secondly, I see a limited novelty of this study. Many decades ago there were studies of photocatalytic activities on different single crystal surfaces. If one removes this part of paper novelty, then there are already published papers for CH₄ coupling on Pd/TiO₂ photocatalyst. While there are some potentially

novel elements in this paper, I do not see urgency or need to publish this paper in this high impact journal.

Reply and actions taken: We thank the reviewer for the comment on our manuscript. However, we cannot agree on the two major points raised by this reviewer. Our rebuttals are as follows.

First, regarding the validity of using nanocrystals to investigate the facet effect, the methodology of using nanocrystals with regular morphologies to gain insight into the facet effect has now been widely accepted in catalysis field (please see for examples: *Chem. Soc. Rev.* **43**, 1543-1574 (2014); *Nat. Energy* **4**, 957-968 (2019); *Science* **371**, 517-521 (2021)). It is known that nanocrystals have controllable exposed facets similar to single crystals, but nanocrystals usually have larger specific surface areas and thus much higher activities than single crystals. Further, nanocrystals of some materials, in particular noble metals and metal oxides, can be synthesized facilely nowadays, owing to the advances in the nanotechnology. These advantages make the nanocrystals highly suitable for the study of the facet-performance relationships. In our present work, we have adopted relatively mature methods to synthesize TiO₂ nanocrystals, and the morphologies and facets of TiO₂ have been carefully characterized. We have drawn useful conclusion on a very unique facet-performance relationship for the photocatalytic NOCM reaction. Many studies have adopted similar methodology in the field of photocatalysis (please see for examples: *J. Am. Chem. Soc.* **136**, 8839-8842 (2014); *ACS Catal.* **6**, 1097-1108 (2016); *Angew. Chem. Int. Ed.* **60**, 6160-6169 (2021); *Chem* **6**, 3038-3053 (2020)).

Second, regarding the novelty of our present work, we would like to emphasize the originality from the following three aspects. **First, we discovered an unusual facet effect.** We discovered surprisingly that the high-energy polar {001} facet of anatase TiO₂, which is generally believed to be a superior facet in photocatalysis (*Nature* **453**, 638-641 (2008)), showed significantly lower activity and selectivity for C₂ formation than the stable {101} facet. Our in-depth experimental and computational work has offered a new insight that the higher reactivity of the {101} facet arises from the higher concentration of •OH radicals in the liquid phase. **Second, we found the crucial role of Pd²⁺ sites on the Pd co-catalyst in C₂ formation.** We found unexpectedly that the fraction of Pd²⁺ on Pd co-catalyst correlated well with the C₂ selectivity, while the fraction of Pd⁰ correlated with the charge separation ability (Fig. 2d). Thus, we propose that the Pd co-catalyst not only accelerates the electron-hole separation as in many photocatalysis systems but also aids the coupling of •CH₃ radicals by adsorbing and enriching them for C–C coupling. **Third, we propose a novel heterogeneous-homogeneous reaction mechanism for photocatalysis.** The heterogeneous-homogeneous mechanism has been found in some thermocatalytic systems such as

oxidative coupling of methane at high temperatures ($>600\text{ }^{\circ}\text{C}$), but similar mechanisms have seldom been reported in photocatalysis. We unveiled that the photogenerated holes on TiO_2 surfaces first activate H_2O , while the $\bullet\text{OH}$ radicals formed in aqueous phase near the surface account for the activation of CH_4 molecules into $\bullet\text{CH}_3$ radicals. Further, the coupling of $\bullet\text{CH}_3$ radicals mainly take place on the Pd^{2+} site on Pd nanoparticle surfaces. Such the heterogeneous-homogeneous mechanism would enable the separation of the reactive oxygen species and the intermediates into different phases, thus alleviating the trade-off problem between activity and selectivity.

REVIEWER COMMENTS

Reviewer #2 (Remarks to the Author):

The manuscript depicts the synthesis of the different exposed facets of TiO₂ and uses them for CH₄ activation with the use of Pd cocatalyst which leads to the formation of the C₂ products by Photocatalytic pathway. The direct CH₄ conversion to value-added C₂ products (ethane or ethylene) has become one of most attractive research goals. Interestingly, the photocatalytic non-oxidative coupling of CH₄ (NOCM) to C₂ products is highly appealing because it additionally produces H₂ without CO₂ emissions and operates under mild reaction conditions. Authors describe the role of 'OH for the activation of methane and comes up with the new kind of heterogeneous-homogeneous reaction mechanism. Although the work is unique in terms of the mechanism depiction and synthetic strategy. I find the journal lacks novelty for the publication in the 'Nature Communication' journal. However, I am not fully convinced with the manuscript due to the following issues:

1. The authors have synthesized different facets of TiO₂ and explored their activity for methane activation. Also, different synthesis strategies can give rise of the different amount of oxygen vacancies, that might give a bigger impact in catalysis. Authors have not commented on this.
2. From the powder-XRD analysis (Figure S3) it is evident that different TiO₂ samples having different peak broadening and the (200) of the TiO₂[001] is intense than the (101), the reason is not explained.
3. In the Raman study the Eg peak of the TiO₂[001] is relatively less intense than the other version of the TiO₂, please explain the reason.
4. Author should provide the powder-XRD with the comparison with Pd simulated pattern.
5. The authors vary the synthetic methods to change the ratio of the Pd oxidation state, but here the PdCl₂ uses as the precursor, but how authors get rid of Cl⁻ from the sample?
6. Authors depicted that the electron-hole separation ability by Transient Photocurrent measurement, please provide the detailed description of the experiment and applied potential, and why the electron-hole separation is a facet dependent property, please explain.
7. Authors should provide any experimental proof for the presence of the methyl radical, and apart from the electron hole separation it is better if author provides any descriptor for depicting the difference the activity for C-C coupling of different TiO₂ that will be very helpful.
8. Any in situ techniques that can validate the theoretical results will be useful for the mechanistic investigations.

Reviewer #3 (Remarks to the Author):

The authors provided the detailed scientific explanation along with relevant references to my queries and improved revised the manuscript accordingly.

I am satisfied with the revised manuscript and recommending for publication.

Reviewer #4 (Remarks to the Author):

In general the author reasonably addressed the issues raised by the reviewers. Although I still am not particularly impressed by novelty of the work but there is no harm in publishing

this work here.

Responses to Reviewers

Response to Reviewer 2

General Comments: The manuscript depicts the synthesis of the different exposed facets of TiO₂ and uses them for CH₄ activation with the use of Pd cocatalyst which leads to the formation of the C₂ products by Photocatalytic pathway. The direct CH₄ conversion to value-added C₂ products (ethane or ethylene) has become one of most attractive research goals. Interestingly, the photocatalytic non-oxidative coupling of CH₄ (NOCM) to C₂ products is highly appealing because it additionally produces H₂ without CO₂ emissions and operates under mild reaction conditions. Authors describe the role of •OH for the activation of methane and comes up with the new kind of heterogeneous-homogeneous reaction mechanism. Although the work is unique in terms of the mechanism depiction and synthetic strategy. I find the journal lacks novelty for the publication in the 'Nature Communication' journal. However, I am not fully convinced with the manuscript due to the following issues:

Reply and actions taken: We appreciate the constructive comments on our manuscript raised by this reviewer.

Because the reviewer mentioned the novelty, here we would like to highlight briefly the novelty of our present work as follows. **First, we discovered an unusual facet effect** that the high-energy polar {001} facet of anatase TiO₂, which is generally believed to be highly reactive in photocatalysis (*Nature* **453**, 638-641 (2008)), showed significantly lower activity and selectivity for C₂ formation than the stable {101} facet. Our experimental and computational studies have offered a new insight that the higher reactivity of the {101} facet arises from the higher concentration of •OH radicals in the liquid phase. **Second, we found unique roles of the Pd co-catalyst in C₂ formation.** Our studies demonstrate that the Pd co-catalyst not only accelerates the electron-hole separation as in many photocatalysis systems but also aids the coupling of •CH₃ radicals by adsorbing and enriching them for C–C coupling, and the co-existence of Pd⁰ and Pd²⁺ is crucial to implement the dual roles of Pd co-catalyst. **Third, we propose a novel heterogeneous-homogeneous reaction mechanism for photocatalysis.** The heterogeneous-homogeneous mechanism has been found in some thermocatalytic systems such as oxidative coupling of methane at high temperatures (>600 °C), but similar mechanisms have seldom been reported in photocatalysis. We have revealed that the photogenerated holes on TiO₂ surfaces first activate H₂O, while the •OH radicals formed in aqueous phase near the surface account for the activation of CH₄ molecules into •CH₃ radicals. Further, the coupling of •CH₃ radicals mainly takes place on the Pd²⁺ site on Pd nanoparticle surfaces. Such the heterogeneous-homogeneous

mechanism would enable the separation of the reactive oxygen species and the intermediates into different phases, thus alleviating the trade-off problem between activity and selectivity.

Our replies to the detailed comments and the corresponding revisions are described as follows.

Comment 1: The authors have synthesized different facets of TiO₂ and explored their activity for methane activation. Also, different synthesis strategies can give rise of the different amount of oxygen vacancies, that might give a bigger impact in catalysis. Authors have not commented on this.

Reply and actions taken: We thank the reviewer for this pertinent comment. Following this comment, we have investigated the effect of the density of oxygen vacancies on the performance of photocatalytic NOCM reaction. The density of oxygen vacancies on the Pd/TiO₂ catalysts during the NOCM reaction has been measured quantitatively by electron titration with thionine acetate (please see Ref. 27: *Chem* **6**, 3038-3053 (2020)). The experimental result has been added in Supplementary Fig. 13 (*please see the next page for quick reference*), which shows that the density of oxygen vacancies decreases in the sequence of Pd/TiO₂-{001} > Pd/TiO₂-{101} > Pd/TiO₂-{100}. This trend is different from the trend for the specific C₂H₆ formation rate and C₂ selectivity, which follows the sequence of Pd/TiO₂-{101} > Pd/TiO₂-{100} > Pd/TiO₂-{001} (*please also see Supplementary Fig. 13*). Therefore, there is no direct correlation between the photocatalytic NOCM performance and the density of oxygen vacancies. In other words, the oxygen vacancies are not the crucial factor determining the catalytic performance in the present system.

A new subsection of “Methods for measuring the density of oxygen vacancies” has been added in “**Supplementary Methods**” section in the revised Supplementary Information (*please see from Page 2-the last paragraph to Page 3-Paragraph 1*). We have also added the following sentence in the revised main text to describe the experimental result mentioned above: “*The density of oxygen vacancies, which might determine the surface reactivity, was measured quantitatively by electron titration²⁷, and it followed a trend of Pd/TiO₂-{001} > Pd/TiO₂-{101} > Pd/TiO₂-{100} (Supplementary Fig. 13). This trend did not correlate well with that for the specific C₂H₆ formation rate and C₂ selectivity*” (*please see from Page 11-Paragraph 2-Line 9 to Page 12-Line 1*).

Supplementary Fig. 13. Density of oxygen vacancies and photocatalytic performance versus exposed facet.

Comment 2: From the powder-XRD analysis (Figure S3) it is evident that different TiO_2 samples having different peak broadening and the (200) of the $\text{TiO}_2\{001\}$ is intense than the (101), the reason is not explained.

Reply and actions taken: We thank the reviewer for this careful comment. It is generally accepted that, the X-ray diffraction line will become broad for a small crystallite, and the smaller the crystal grain, the broader the X-ray diffraction band (please see for examples: *Science* **380**, 1174-1178 (2023); *ACS Catal.* **11**, 9022-9033 (2021)). Our TEM results showed that the average sizes roughly followed the trend of $\text{TiO}_2\text{-}\{101\} > \text{TiO}_2\text{-}\{001\}$ (the thickness is smaller) $> \text{TiO}_2\text{-}\{100\}$, despite of the irregular morphologies of the samples (Supplementary Fig. 1). Therefore, we believe that the different peak broadening of different samples mainly arises from the different average sizes of samples, and the $\text{TiO}_2\text{-}\{100\}$ sample with the smallest average size exhibited the broadest diffraction peaks.

The (101) peak of TiO_2 is stronger than the (200) peak, attributing to the high multiplicity of the {101} facet (please see Fujishima, A., Hashimoto, K. & Watanabe, T. Titanium Dioxide. *RSC Materials Monographs*, 1999). In other words, there are more equivalent {101} facets in the crystallite interacting with the incident X-ray, resulting in a stronger (101) signal. As compared to $\text{TiO}_2\text{-}\{101\}$ and $\text{TiO}_2\text{-}\{100\}$, the (200) peak of the $\text{TiO}_2\text{-}\{001\}$ sample is more intense. This is because the $\text{TiO}_2\text{-}\{001\}$ sample (TiO_2 nanosheet) has the smallest thickness in the [001] direction and the largest side length in the [100] direction.

We have added the following sentences in the revised main text to describe the

result described above: “*The difference in the broadening of XRD peaks probably arises from the difference in the average sizes of TiO₂ nanocrystals. Compared to TiO₂-{101} and TiO₂-{100}, the TiO₂-{001} sample showed relatively stronger (200) and weaker (004) diffraction peaks (Supplementary Fig. 3a), in agreement with the TEM result that the TiO₂-{001} sample has the largest side length in the [100] direction and the smallest thickness in the [001] direction³¹*” (please see Page 6, Lines 7-11).

Comment 3: In the Raman study the E_g peak of the TiO₂[001] is relatively less intense than the other version of the TiO₂, please explain the reason.

Reply and actions taken: Raman bands originate from the vibrations of chemical bonds and depend on the exposed facet of TiO₂ (*J. Phys. Chem. C* **116**, 7515-7519 (2012)). The E_g bands for anatase TiO₂ at 144 and 636 cm⁻¹ are attributable to the symmetric stretching vibrations of O–Ti–O. The {001} facet of TiO₂ only contains the unsaturated 5c-Ti and 2c-O bonding modes (*Nature* **453**, 638-641 (2008)), and thus the TiO₂-{001} surface has fewer symmetric stretching vibration modes of O–Ti–O. In contrast, the bonding modes on the surfaces of TiO₂-{101} and TiO₂-{100} contain both unsaturated (5c-Ti and 2c-O) and saturated (6c-Ti and 3c-O) bonding modes (*J. Mater. Chem. A* **1**, 10532-10537 (2013)), and thus these two surfaces have more symmetric stretching vibration modes of O–Ti–O. Therefore, the TiO₂-{001} sample shows weaker E_g bands than the TiO₂-{101} and TiO₂-{100} samples.

We have added the following sentence in the revised main text to describe the discussion above: “*The E_g bands at 144 and 636 cm⁻¹ in Raman spectra are attributable to the symmetric stretching vibration of O–Ti–O of TiO₂ (Ref.³¹), and the weaker E_g bands for TiO₂-{001} are consistent with the fact that this sample exposes a higher fraction of {001} facet with fewer symmetric stretching vibration modes of O–Ti–O (Supplementary Fig. 3b)*” (please see Page 6, Lines 11-15).

Comment 4: Author should provide the powder-XRD with the comparison with Pd simulated pattern.

Reply and actions taken: We thank the reviewer for this reminding. We have added Pd simulated pattern (in blue line) in Supplementary Fig. 3f in the revised Supplementary Information. Now, it can be seen more clearly that the crystalline structures of the three TiO₂ samples did not change and the diffraction lines of Pd could not be observed after loading Pd (please also see Supplementary Fig. 3f displayed in the next page for quick reference).

Supplementary Fig. 3 | Characterizations of crystalline and surface structures. **a**, XRD patterns. **b**, Raman spectra. **c**, XPS survey spectra. **d**, Ti 2p XPS spectra. **e**, O 1s XPS spectra. **f**, XRD patterns. The signal of C 1s in XPS survey spectra originates from the contaminant carbon, which is typically used as the standard for calibration of binding energy values.

Comment 5: The authors vary the synthetic methods to change the ratio of the Pd oxidation state, but here the PdCl₂ uses as the precursor, but how authors get rid of Cl⁻ from the sample?

Reply and actions taken: We thank the reviewer for this constructive comment. We have measured the surface content of the remaining chlorine by XPS for the Pd/TiO₂-{101} sample and found that some Cl⁻ residues indeed remain on the sample. The result has been added in Supplementary Fig. 7 in the revised Supplementary Information (*please also see the next page for quick reference*). To investigate the effect of the remaining Cl⁻ on the photocatalytic performance, we have prepared Cl⁻-free Pd/TiO₂-{101} catalyst by the same adsorption-reduction method except for using Pd(NO₃)₂ as the Pd precursor instead of PdCl₂. The comparison of the photocatalytic NOCM performances between the original Pd/TiO₂-{101} (PdCl₂ as the Pd precursor) and the Cl⁻-free Pd/TiO₂-{101} (Pd(NO₃)₂ as the Pd precursor) catalysts shows that these two catalysts have similar CH₄ conversions and C₂ selectivities as well as the product formation rates (*please see Supplementary Table 4 in the revised Supplementary Information, and please also see the next page for quick reference*). Therefore, the use of PdCl₂ as the precursor may cause Cl⁻ residues on catalyst surfaces, but the effect of Cl⁻ residues on photocatalytic NOCM performances can be ignored.

We have added the following sentence in the revised main text to describe the result

and discussion described above: “We prepared Cl^- -free $\text{Pd/TiO}_2\text{-}\{101\}$ catalyst by using $\text{Pd}(\text{NO}_3)_2$ as the Pd precursor instead of PdCl_2 , which was typically used in the present work, and the XPS measurement confirmed the absence of chlorine on the surface of this catalyst (Supplementary Fig. 7). The $\text{Pd/TiO}_2\text{-}\{101\}$ and the Cl^- -free $\text{Pd/TiO}_2\text{-}\{101\}$ catalysts exhibited similar performances for photocatalytic NOCM (Supplementary Table 4), excluding the effect of Cl^- residues” (please see Page 9, Paragraph 2, Lines 9-14).

Supplementary Fig. 7 | Cl 2p XPS spectra for Pd-loaded $\text{TiO}_2\text{-}\{101\}$ catalysts. PdCl_2 or $\text{Pd}(\text{NO}_3)_2$ was used as the Pd precursor.

Supplementary Table 4 | Photocatalytic NOCM performances

Catalyst	Product amount (μmol)				CH_4 conversion ^a (%)	C_2 selectivity ^b (%)	C_2 yield ^c (%)
	C_2H_6	C_2H_4	CO_2	H_2			
$\text{Pd/TiO}_2\text{-}\{101\}$	10	0.16	4.8	42	1.3	81	1.1
$\text{Pd/TiO}_2\text{-}\{100\}$	2.8	0	2.5	13	0.40	69	0.27
$\text{Pd/TiO}_2\text{-}\{001\}$	1.0	0	4.2	6.4	0.30	31	0.094
Pd/P25	8.1	0.26	13	61	1.4	56	0.77
$\text{Pd/TiO}_2\text{-}\{101\}$^d	11	0	5.3	40	1.4	81	1.1
$\text{TiO}_2\text{-}\{101\}$ + $\text{Pd/Al}_2\text{O}_3$	2.3	0	4.0	12	0.43	53	0.23

Reaction conditions: catalyst, 20 mg; water, 50 mL; CH_4 , 45 mL (2009 μmol); light source, 300 W Xe lamp ($\lambda = 320\text{-}780$ nm); irradiation time, 4 h.

^a CH_4 conversion = $[2 \times n(\text{C}_2\text{H}_6) + 2 \times n(\text{C}_2\text{H}_4) + n(\text{CO}_2)] / n(\text{CH}_4)$

^b C_2 selectivity = $[2 \times n(\text{C}_2\text{H}_6) + 2 \times n(\text{C}_2\text{H}_4)] / [2 \times n(\text{C}_2\text{H}_6) + 2 \times n(\text{C}_2\text{H}_4) + n(\text{CO}_2)]$

^c C_2 yield = CH_4 conversion \times C_2 selectivity

^d $\text{Pd}(\text{NO}_3)_2$ as the Pd precursor

Comment 6: Authors depicted that the electron-hole separation ability by Transient Photocurrent measurement, please provide the detailed description of the experiment and applied potential, and why the electron-hole separation is a facet dependent property, please explain.

Reply and actions taken: We thank the reviewer for this pertinent comment. First, we have added a new subsection of “Transient photocurrent response measurements” in the “Supplementary Methods” section in the revised Supplementary Information (*please see Page 3, Paragraph 2 in the revised Supplementary Information*). Second, as the reviewer pointed out, the result obtained from the transient photocurrent response measurement shows that the charge-separation efficiency is facet dependent and decreases in the sequence of Pd/TiO₂-{101} ≈ Pd/TiO₂-{001} > Pd/TiO₂-{100} (inset of Fig. 3a). Regarding why the electron-hole separation is face-dependent, we speculate that the differences in the surface energy (*ACS Catal.* **12**, 6457-6463 (2022)) and in the anisotropy (*J. Am. Chem. Soc.* **136**, 8839-8842 (2014); *Appl. Catal. B: Environ.* **198**, 286-294 (2016)) among different exposed TiO₂ facets are the major reasons for the difference in the charge separation efficiency. However, the C₂H₆ formation rate and the C₂ selectivity, which decrease in the sequence of Pd/TiO₂-{101} > Pd/TiO₂-{100} > Pd/TiO₂-{001} (Fig. 2a), do not correlate with the electron-hole separation ability. In other words, the charge separation ability is not a crucial factor determining the photocatalytic NOCM performance. Actually, we already pointed out that “*it is not the shape-based parameters, which may affect the light absorption, the electron-hole separation, and the surface area or the density of oxygen vacancies, but the exposed facet-based factor (not just the surface energy) plays a determining role in the TiO₂-catalysed NOCM*” (*please see Page 12, Lines 3-6 in the revised manuscript*). Therefore, we have not made further discussion on the reason for why the charge separation ability is facet dependent in the revised manuscript.

Comment 7: Authors should provide any experimental proof for the presence of the methyl radical, and apart from the electron hole separation it is better if author provides any descriptor for depicting the difference the activity for C-C coupling of different TiO₂ that will be very helpful.

Reply and actions taken: We appreciate this constructive comment from the reviewer.

First, we would like to point out that, unlike the insight in many photocatalytic systems, the electron-hole separation ability is not the determining factor or the descriptor in the present system for the photocatalytic NOCM. As mentioned above in the response to Comment 6 raised by this reviewer, our studies reveal that the C₂H₆ formation rate and the C₂ selectivity, both of which decrease in the sequence of Pd/TiO₂-{101} > Pd/TiO₂-{100} > Pd/TiO₂-{001} (Fig. 2a), do not correlate with the

electron-hole separation ability (inset of Fig. 3a).

The experimental and theoretical studies in the present work demonstrate that the formation energy of the $\bullet\text{OH}$ radicals in liquid phase and the surface fraction of Pd^{2+} correlate well with the C_2 formation rate and the C_2 selectivity, respectively (Fig. 3c and Fig. 2d). Actually, one important insight obtained from the present work is that the concentration of $\bullet\text{OH}$ radicals in the liquid phase determines the C_2 formation rate (Fig. 3c). This can explain our discovery that the most stable TiO_2 {101} facet shows the best performance for the photocatalytic NOCM, because the concentration of $\bullet\text{OH}$ radicals in the liquid phase for this sample is the highest (Fig. 3c). In contrast, the high-energy surface of TiO_2 {001} has high adsorption energy of $\bullet\text{OH}$ radicals, leading to difficulty in generating $\bullet\text{OH}$ radicals in the liquid phase. These two factors are also related to the main points of the mechanism (Fig. 4c), i.e., $\bullet\text{OH}$ radicals in the liquid phase responsible for CH_4 activation and Pd^{2+} responsible for the coupling of $\text{CH}_3\bullet$ radicals. Thus, these two factors can be used as descriptors for the activity and selectivity in photocatalytic NOCM. To make our points more easily understood, we have added the following sentence in the “**Conclusion**” section in the revised manuscript to highlight the importance of the two descriptors: “*The formation energy of $\bullet\text{OH}$ radicals in the liquid phase and the surface fraction of Pd^{2+} are two key descriptors in the present system*” (please see Page 16, Paragraph 2, Lines 6-8).

Following the comment raised by this reviewer, we have attempted to measure methyl radicals ($\bullet\text{CH}_3$) by *in situ* electron spin resonance (ESR) spectroscopic technology using 5,5-dimethyl-1-pyrroline N-oxide (DMPO) as a radical trapping agent, but no signal corresponding to DMPO- CH_3 spin adduct could be detected. We speculate that the concentration of $\bullet\text{CH}_3$ is too low in the reaction solution to reach the detection limit of the instrument. We could detect $\bullet\text{OH}$ radicals because of the higher concentration of $\bullet\text{OH}$ in liquid phase. Nevertheless, it should be noted that our DFT calculations reveal that the $\bullet\text{OH}$ radical is capable of activating CH_4 via H abstraction to generate $\bullet\text{CH}_3$ (Supplementary Fig. 18). The formation of C_2H_6 as the predominant C_2 product also supports $\bullet\text{CH}_3$ as the intermediate. Further, the activation of CH_4 to $\bullet\text{CH}_3$, followed by subsequent transformations of $\bullet\text{CH}_3$ radicals, has been widely accepted in photocatalysis field (please see for examples: *CCS Chem.* **5**, 30-54 (2022); *Chem Catal.* **3**, 100437 (2023)). Because the formation of $\bullet\text{CH}_3$ as the intermediate is not a unique point of the present work, we have not made further characterization of $\bullet\text{CH}_3$. This will not spoil the story and the rigorousness of the present manuscript.

Comment 8: Any *in situ* techniques that can validate the theoretical results will be useful for the mechanistic investigations.

Reply and actions taken: We thank the reviewer for this constructive comment. Our

density functional theory (DFT) calculations have mainly offered the following insights: (1) The liquid-phase $\bullet\text{OH}$ radicals could be formed more facily over the $\text{TiO}_2\text{-}\{101\}$ catalyst as compared to the $\text{TiO}_2\text{-}\{100\}$ and $\text{TiO}_2\text{-}\{001\}$ catalysts; (2) The $\bullet\text{OH}$ radical are capable of activating CH_4 via H abstraction to generate $\bullet\text{CH}_3$; (3) The Pd nanoparticles can function for catching $\bullet\text{CH}_3$ radicals from liquid phase and aiding the coupling of $\bullet\text{CH}_3$ radicals to form C_2 compounds, and Pd^{2+} sites on Pd nanoparticles play crucial roles.

The above theoretical results have been validated by the following experiments. **First**, the generation of $\bullet\text{OH}$ radicals has been confirmed by *in situ* ESR; the ESR result shows that the concentration of $\bullet\text{OH}$ radicals in liquid phase decreases in the sequence of $\text{Pd/TiO}_2\text{-}\{101\} > \text{Pd/TiO}_2\text{-}\{100\} > \text{Pd/TiO}_2\text{-}\{001\}$ (Fig. 3b). The concentration and the ratio of the $\bullet\text{OH}$ radicals in the liquid phase have further been quantified by *in situ* fluorescence measurements (Figs. 3c and 3d), and both follow the trend of $\text{Pd/TiO}_2\text{-}\{101\} > \text{Pd/TiO}_2\text{-}\{100\} > \text{Pd/TiO}_2\text{-}\{001\}$. **Second**, when terephthalic acid (25 μmol), a scavenger of $\bullet\text{OH}$ radicals, was added to the photocatalytic system with the $\text{Pd/TiO}_2\text{-}\{101\}$ catalyst, the amounts of C_2H_6 , C_2H_4 , and CO_2 formed after 2 h of irradiation decreased drastically from 6.4, 0.16, and 3.1 μmol to 0.12, 0, and 0.28 μmol , respectively, indicating that the $\bullet\text{OH}$ radical participates in the conversion of CH_4 . We have tried to measure $\bullet\text{CH}_3$ by *in situ* ESR spectroscopy using DMPO as a radical trapping agent, but no signal corresponding to DMPO- CH_3 spin adduct has been detected, probably because the concentration of $\bullet\text{CH}_3$ in the liquid phase is too low to reach the detection limit of the instrument. **Third**, we have designed an experiment to confirm the role of Pd sites in aiding the coupling of $\bullet\text{CH}_3$ to C_2H_6 . In brief, when the $\text{TiO}_2\text{-}\{101\}$ catalyst was physically mixed with $\text{Pd/Al}_2\text{O}_3$ (itself inactivity for photocatalytic NOCM), C_2H_6 with 53% selectivity could also be obtained (Supplementary Table 4, *please also see the next page for quick reference*). This means that the interfacial contact between Pd and $\text{TiO}_2\text{-}\{101\}$ is not necessary for NOCM, supporting the insight from the DFT calculation that Pd^{2+} sites function for catching $\bullet\text{CH}_3$ radicals from liquid phase and aid the coupling of $\bullet\text{CH}_3$ radicals.

The experimental results regarding the measurements of $\bullet\text{OH}$ radicals as well as their participation in the photocatalytic NOCM were already included in our previous version of manuscript. In the revised manuscript, we have added the following sentences to describe the experimental result regarding the use of mixture of $\text{TiO}_2\text{-}\{101\}$ and $\text{Pd/Al}_2\text{O}_3$: “*Experimentally, when a physical mixture of the $\text{TiO}_2\text{-}\{101\}$ sample and a $\text{Pd/Al}_2\text{O}_3$ catalyst, which itself was inactive for photocatalytic NOCM, was used instead of the $\text{Pd/TiO}_2\text{-}\{101\}$ catalyst, CH_4 could be converted into C_2H_6 with 53% selectivity (Supplementary Table 4). This supports the result obtained from DFT calculations that the Pd co-catalyst with Pd^{2+} contributes to aiding the coupling by*

trapping •CH₃ radicals from the liquid phase” (please see Page 15, Lines 9-14).

Supplementary Table 4 | Photocatalytic NOCM performances

Catalyst	Product amount (μmol)				CH ₄ conversion ^a (%)	C ₂ selectivity ^b (%)	C ₂ yield ^c (%)
	C ₂ H ₆	C ₂ H ₄	CO ₂	H ₂			
Pd/TiO ₂ -{101}	10	0.16	4.8	42	1.3	81	1.1
Pd/TiO ₂ -{100}	2.8	0	2.5	13	0.40	69	0.27
Pd/TiO ₂ -{001}	1.0	0	4.2	6.4	0.30	31	0.094
Pd/P25	8.1	0.26	13	61	1.4	56	0.77
Pd/TiO ₂ -{101} ^d	11	0	5.3	40	1.4	81	1.1
TiO ₂ -{101} + Pd/Al ₂ O ₃	2.3	0	4.0	12	0.43	53	0.23

Reaction conditions: catalyst, 20 mg; water, 50 mL; CH₄, 45 mL (2009 μmol); light source, 300 W Xe lamp (λ = 320-780 nm); irradiation time, 4 h.

^a CH₄ conversion = $[2 \times n(\text{C}_2\text{H}_6) + 2 \times n(\text{C}_2\text{H}_4) + n(\text{CO}_2)] / n(\text{CH}_4)$

^b C₂ selectivity = $[2 \times n(\text{C}_2\text{H}_6) + 2 \times n(\text{C}_2\text{H}_4)] / [2 \times n(\text{C}_2\text{H}_6) + 2 \times n(\text{C}_2\text{H}_4) + n(\text{CO}_2)]$

^c C₂ yield = CH₄ conversion × C₂ selectivity

^d Pd(NO₃)₂ as the Pd precursor

Response to Reviewer 3

General Comments: The authors provided the detailed scientific explanation along with relevant references to my queries and improved revised the manuscript accordingly. I am satisfied with the revised manuscript and recommending for publication.

Reply and actions taken: We appreciate the kind comments on our manuscript raised by reviewer 3.

Response to Reviewer 4

General Comments: In general, the author reasonably addressed the issues raised by the reviewers. Although I still am not particularly impressed by novelty of the work but there is no harm in publishing this work here.

Reply and actions taken: We appreciate the kind comments on our manuscript raised by reviewer 4.

REVIEWERS' COMMENTS

Reviewer #2 (Remarks to the Author):

Authors have responded to the comments with appropriate revisions in the manuscript. It can be accepted for publication Nature Communications.

Responses to Reviewer 2

General Comments: Authors have responded to the comments with appropriate revisions in the manuscript. It can be accepted for publication Nature Communications.

Reply and actions taken: We thank the reviewer for the kind comment. There are no revision requests now by the reviewer.